# RETHINKING NONLINEAR DYNAMICS IN DEEP TIME SERIES MODELS

## ABSTRACT

Deep learning models have achieved remarkable success in modeling complex time series data, yet their black-box nature limits interpretability and explicit representation of intrinsic dynamic structures such as nonlinear interactions and memory effects. Observing the inherent compatibility of Volterra series' polynomial integral kernels with GPU-accelerated deep learning frameworks, we propose the Discrete Volterra Network (DiVo), a novel deep learning model family integrating Volterra series to explicitly learn dynamic characteristics from time series data. Specifically, DiVo computes discrete Volterra coefficient matrices via polynomial expansions, converting nonlinear time series modeling into linear polynomial coefficient learning. To address practical challenges, we introduce adaptive channel selection to remove strict dependence on time-invariant sequences, and propose a redundancy-aware sparsification strategy that combines fixed masking of Volterra features with sparsified low-rank decomposition to eliminate redundancy in both the feature and parameter spaces, yielding a compact model representation.Extensive experiments on diverse real-world datasets show DiVo significantly outperforms traditional deep models in prediction accuracy, interpretability, and parameter efficiency.

## 1 INTRODUCTION

Over the past decade, deep learning has become a dominant approach for modeling complex time series data, due to its strong capability to learn directly from large-scale data. However, current deep learning models typically behave as black-boxes, composed of linear and nonlinear functions, without explicitly capturing intrinsic dynamic structures (such as nonlinear interactions and memory effects) inherent in real-world nonlinear systems. This limits their interpretability and theoretical capability in precisely modeling highly nonlinear dynamic systems.

The Volterra series has long been regarded as a powerful and theoretically grounded tool for modeling nonlinear dynamic systems. Compared to classical methods such as Wiener, Hammerstein, or NARX models, which often rely on fixed structural assumptions or low-order approximations, Volterra provides a more flexible and expressive formulation by expanding system dynamics into a series of high-order polynomial interactions over input history. In principle, the convolutional structure of the Volterra expansion aligns well with tensor-based computations, making it seemingly suitable for modern parallel computing environments.

However, despite these strengths, the original Volterra formulation is fundamentally incompatible with large-scale deep learning for three key reasons: (1) it relies on continuous multi-dimensional integrals, which cannot be directly represented or optimized in standard deep learning frameworks; (2) it assumes strict time-invariance, which rarely holds in noisy, real-world sequences; and (3) it suffers from exponential parameter growth, making high-order modeling intractable. These limitations render direct application of the Volterra series impractical, and motivate the need for a principled reparameterization that transforms its theoretical advantages into a usable, scalable, and learnable form.

To overcome the first limitation, namely the incompatibility of continuous Volterra integrals with standard deep learning frameworks, we introduce a **Discrete Volterra Reparameterization** that reformulates the original integral representation into a fully discrete, learnable, and structured form. Concretely, we construct a set of **Volterra Feature Matrices** by applying Kronecker-powered polynomial expansions to the input time series, encoding high-order nonlinear interactions over time.

Correspondingly, each high-order Volterra kernel is reparameterized as a learnable coefficient matrix that operates directly on these features. This transformation yields a clean bilinear form between features and parameters, converting the nonlinear integral modeling problem into a linear coefficient estimation task. As illustrated in Figure 1, we refer to the collection of these coefficient matrices as the **Discrete Volterra Coefficients Matrix**, which encapsulates the system's nonlinear dynamics in a structured, interpretable, and GPU-compatible format.

To address the remaining two challenges, including the assumption of strict time-invariance and the exponential growth of parameters, we design the **Discrete Volterra Network (DiVo)**. This deep time series modeling framework builds upon the structured reparameterization and organizes the prediction process as the application of multiple coefficient matrices across dynamically changing inputs. To relax the constraint of time-invariant kernels, DiVo introduces a multi-channel mechanism that allows different coefficient matrices to operate on different temporal contexts. This design enables the model to adapt to local variations in the data while maintaining global structural consistency. To further reduce parameter redundancy, we apply a redundancy-aware sparsification strategy that combines fixed feature masking and sparsified low-rank decomposition. These designs promote compactness, improve generalization, and support efficient training. The resulting architecture retains the expressive power of Volterra modeling while achieving scalability, interpretability, and strong empirical performance on real-world forecasting tasks, as well as synthetic chaotic systems.

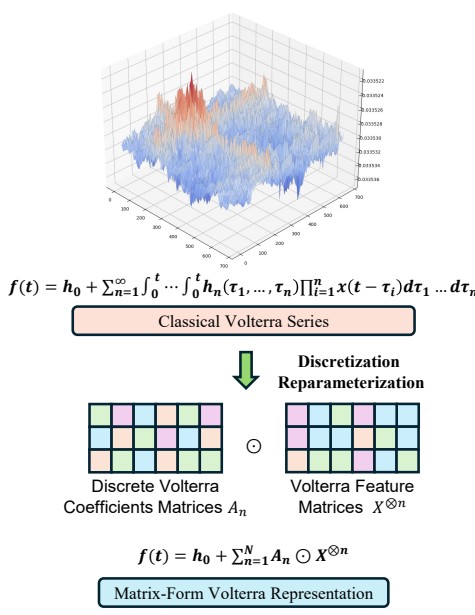

$$f(t) = h_0 + \sum_{n=1}^{\infty} \int_0^t \cdots \int_0^t h_n(\tau_1, \dots, \tau_n) \prod_{i=1}^n x(t - \tau_i) d\tau_1 \dots d\tau_n$$

Classical Volterra Series

**Discretization Reparameterization**

Discrete Volterra Coefficients Matrices $A_n$ $\odot$ Volterra Feature Matrices $X^{\otimes n}$

$$f(t) = h_0 + \sum_{n=1}^{N} A_n \odot X^{\otimes n}$$

Matrix-Form Volterra Representation

Figure 1: Discrete Volterra Reparameterization forming the Discrete Volterra Coefficients Matrix.

**Simplicity** DiVo replaces traditional neural network nonlinearities with structured high-order polynomial expansions, eliminating the need for hand-crafted activation functions or deep stacking. The prediction module only requires a lightweight linear layer to estimate the coefficients of each polynomial order, simplifying model design (For detailed explanations, refer to Section 4, and for experimental results, see Section 6.4, Appendix G.1, Appendix G.2 and Appendix G.3 ).

**Performance** Experiments on diverse real-world forecasting tasks confirm that DiVo achieves strong predictive accuracy while maintaining model compactness and interpretability, with an average improvement of 16.4% over classical deep time series models. (The experimental results in Section 6.2, Section 6.3, Appendix G.4 and Appendix G.5 support our conclusions.)

**Interpretability** By explicitly modeling high-order temporal interactions through discrete Volterra features and structured coefficient matrices, DiVo enables parameter-level interpretability, revealing how different input variables interact across time. This interpretability is demonstrated in our weather forecasting experiments, where the learned coefficients capture meaningful cross-variable dependencies (Refer to Appendix E and case study in Section 6.5, Appendix G.6, Appendix G.7).

## 2 RELATED WORK

The existing work related on time series modeling spans into two broad categories: deep time series models and dynamical methods. **Deep time series models** include autoregressive models like ARIMA Ho & Xie (1998), RNN variants (LSTM Graves & Graves (2012), GRU Dey & Salem (2017)), fully connected networks (TSMixer Chen et al. (2023), WPMixer Murad et al. (2025), TimeXer Wang et al. (2024)) with strong approximation capabilities but limited interpretability Goodfellow et al. (2016), and transformers (Informer Zhou et al. (2021), Autoformer Wu et al. (2021), FEDformer Zhou et al. (2022b), Crossformer Zhang & Yan (2023)) leveraging attention mechanisms Vaswani (2017) while struggling with interpret nonlinear dynamics. Convolutional approaches like

ModernTCN Luo & Wang (2024) and attention-based models such as DeformTime Shu & Lampos (2024) have shown improvements in capturing temporal patterns. Linear models like DLinear Zeng et al. (2023) demonstrate effectiveness through decomposition strategies, while parameter-efficient methods like SparseTSF Lin et al. (2025), CycleNet Lin et al. (2024), and FITS Xu et al. (2023) achieve strong performance with minimal parameters. Other models like Neural ODEs Chen et al. (2018) face long-term dependency challenges, and KAN Liu et al. (2024) encounters scalability issues. **Dynamical methods** employ Volterra series Zheng & Zafiriou (1996); Wray & Green (1994); Cheng et al. (2017); Borys (2018) and hybrid approaches combining Volterra series with neural architectures Oh & Pedrycz (2002); Chrysos et al. (2022), though complexity grows exponentially with system order. A comprehensive review of related work is provided in Appendix B.

## 3 PRELIMINARIES

In this section, we approach from the perspective of dynamic nonlinear systems to introduce the Volterra series, introduce its discrete and learnable reformulation, and redefine time series prediction as a matrix-form coefficient learning problem.

**Volterra Series for Nonlinear Dynamics** A commonly used nonlinear dynamic system can be described as follows Vidyasagar (2002):

$$x(t + \delta t) = \int_0^t f(x(\tau))d\tau \tag{1}$$

where $f(x(\tau))$ is a nonlinear function, $x(\tau)$ is the state variable, and $t$ is time.

According to Volterra series theory, under the assumption of time-invariant interactions, the nonlinear function $f(x(\tau))$ can be approximated by a series of polynomial integral expansions:

$$f(t) = h_0 + \sum_{n=1}^{\infty} \int_0^t \cdots \int_0^t h_n(\tau_1, \ldots, \tau_n) \prod_{i=1}^n x(t - \tau_i)d\tau_1 \ldots d\tau_n \tag{2}$$

where $n$ denotes the polynomial order, and $h_n(\tau_1, \ldots, \tau_n)$ are the Volterra kernels that characterize nonlinear interactions and memory effects.

**Discrete Reparameterization of Volterra Series**

In real-world applications, time series are finite and discrete. Therefore, we discretize the integral form in Equation 2 as:

$$f(t) = h_0 + \sum_{i_1=1}^{N} \sum_{i_2=1}^{N} \cdots \sum_{i_n=1}^{N} H_n(i_1, \cdots, i_n)\left(X(t - i_1) \otimes X(t - i_2) \otimes \cdots \otimes X(t - i_n)\right) \tag{3}$$

where $N$ is the window length, $\otimes$ denotes the Kronecker product, and $H_n(i_1, \cdots, i_n)$ are the discretized kernel coefficients.

We observe that the nested summation preserves the structural dimensionality of the coefficients. Thus, we reformulate the expression by introducing the concept of **Volterra Feature** $X^{\otimes n}$, the $n$-th order Kronecker power of the input window, and define a learnable **Discrete Volterra Coefficients Matrix** $A^n$ to encode the associated parameters. Consequently, Equation 2 can be rewritten in a fully discrete, learnable form:

$$\sum_{i_1, \cdots, i_n=1}^{N} H_n(i_1, \cdots, i_n)\left(\bigotimes_{j=1}^n X(t - i_j)\right) = A^n \odot X^{\otimes n} \tag{4}$$

where $\odot$ denotes the Hadamard product.

**Definition 1** (Discrete Volterra Reparameterization). Given an input time series window $X \in \mathbb{R}^{N \times d}$, we define the **Volterra Feature of order** $n$ as the Kronecker power $X^{\otimes n} \in \mathbb{R}^{d^n}$, and the corresponding learnable **Discrete Volterra Coefficient Matrix** $A^n \in \mathbb{R}^{d^n}$. The **Discrete Volterra Reparameterization** is the transformation:

$$f(t) = h_0 + \sum_{n=1}^{k} A^n \odot X^{\otimes n} \tag{5}$$

which replaces the original integral Volterra kernels with a structured, fully discrete, and learnable formulation, enabling scalable training and direct integration into deep learning architectures.

**Truncation and Multi-Channel Extension** In practice, we truncate the maximum order at $k$ to avoid the combinatorial growth of parameters. Additionally, since real-world time series often violate strict time-invariance, we extend the representation by introducing a set of $l$ coefficient matrix groups, each weighted by a learnable scalar $\lambda_j$, leading to:

$$f(t) = h_0 + \sum_{j=1}^{l} \lambda_j \left( A_j + \sum_{n=1}^{k} A_j^n \odot X^{\otimes n} \right) \tag{6}$$

where $A_j^n$ denotes the $n$-th order coefficient matrix in the $j$-th channel, and $A_j$ is the channel-specific bias. Appendix C provides detailed derivations for this equation. Based on the above derivation, we redefine the time series prediction problem:

**Problem Definition** Given a univariate or multivariate historical time series $\mathcal{X}_o = \{x_1, x_2, \ldots, x_O\}$, $x_t \in \mathbb{R}^d$, our goal is to predict its future values $\hat{x}_{O+1}, \ldots, \hat{x}_{O+H}$ over a fixed horizon $H$. Formally, we aim to learn a function $f(\cdot; \theta)$ such that:

$$[\hat{x}_{O+1}, \ldots, \hat{x}_{O+H}] = f(x_1, x_2, \ldots, x_O; \theta) \tag{7}$$

While traditional time series models adopt RNNs or attention-based networks as $f(\cdot)$, we define it using the matrix-form polynomial expansion in Equation 6, and optimize the discrete Volterra coefficient matrices $\mathcal{A}^k = \{A_1^k, \ldots, A_l^k\}$ in a structured and scalable way.

## 4    DISCRETE VOLTERRA NETWORK

In this section, we introduce the proposed Discrete Volterra Network in detail. The Discrete Volterra Network consists of two parts as shown in Figure 2. First, in the Volterra feature construction module, we compute the Kronecker power representation from the input data. Then, we use a concise multi-channel linear layer to transform the reconstructed Kronecker power representation into predictions.

**Volterra Feature Construction Module** We introduce a Volterra Feature Construction Module to compute high-order polynomial features from the historical input sequence. Given an input sequence $\mathcal{X}_o = \{x_0, x_1, \ldots, x_O\} \in \mathbb{R}^{O \times d}$, we first normalize each time step using the global statistics $\mu$ and $\sigma$, producing a normalized sequence $\tilde{x}_t = (x_t - \mu)/\sigma$. We then flatten the normalized sequence across the observation window into a single vector $\tilde{X}_h \in \mathbb{R}^{Od}$, which serves as the base input for polynomial feature construction. Anaysis of robustness to input perturbations in Appendix D.

To capture high-order nonlinear dependencies, we construct the Volterra feature set $\mathbf{X} = \{X^{\otimes 1}, X^{\otimes 2}, \ldots, X^{\otimes k}\}$, where each $X^{\otimes i} \in \mathbb{R}^{(Od)^i}$ is the $i$-th Kronecker power of $\tilde{X}_h$. These features encode increasingly complex interactions among the input dimensions and form the input to the coefficient estimation module.

**Multi-Channel Linear Transformation Layer**

After computing the Volterra feature, we further design a multi-channel linear transformation layer to capture the nonlinear relationships in equation 6. This layer consists of two key stages: (1) polynomial-integral modeling within each channel and (2) linear aggregation across all channels.

Given the Volterra feature $\mathbf{X}$ computed from the historical sequence, each channel $k$ independently models the nonlinear interactions and memory effects of the system.

In each channel, we set up a learnable Discrete Volterra Coefficients Matrices $\mathbf{A}_k = A_1, A_2, \cdots, A_k$ to capture nonlinear relationships. Specifically, for the output $s_j$ of channel $j$, the calculation process is as follows:

$$s_j = A_j + \sum_{n=1}^{k} A_j^n \odot X^{\otimes n} \tag{8}$$

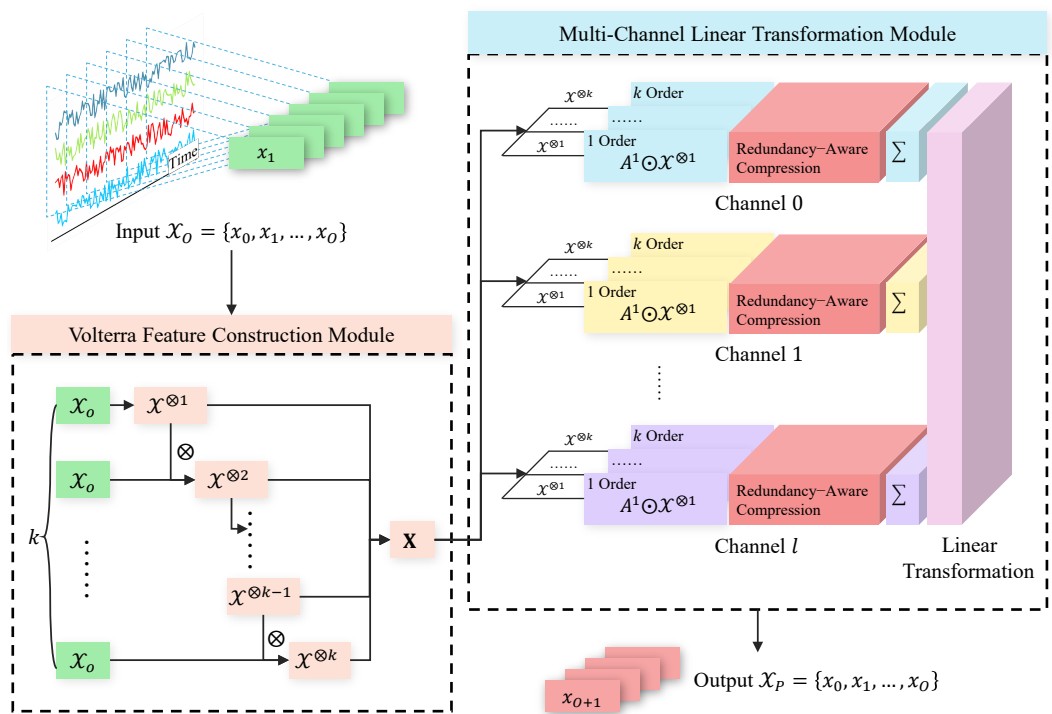

Figure 2: Overview of Discrete Volterra Network.

where $A_j$ is the constant term, $A_j^n$ is the learnable coefficient matrix, and $X^{\otimes n}$ is the Volterra feature. After obtaining the output of each channel, we use a linear transformation layer to aggregate them together and project the output to the original space:

$$X(t+1) = WS^T(t) + B \tag{9}$$

$$S(t) = [s_0(t), s_1(t), \ldots, s_l(t)] \in \mathbb{R}^{l \cdot d} \tag{10}$$

where l is the number of channels, d is the dimension of the time series. $W \in \mathbb{R}^{d \times Kr}$ and $b \in \mathbb{R}^d$ are learnable weight matrix and bias, and $d$ represents the dimension of the target time series.

In summary, the multi-channel linear transformation layer is the neural network implementation of equation 6, which captures complex nonlinear dynamics and long-term dependencies in time series data through multiple independent polynomial-integral channels.

**Loss Function** We choose the Mean Squared Error (MSE) as the loss function to evaluate the prediction performance of DiVo and optimize the network parameters through the gradient descent algorithm.

The implementation of DiVo does not rely on any activation functions, thus avoiding the non-linearity introduced by activation functions. The architecture consisting of only linear layers avoids dependence on specific network structures. Meanwhile, the introduction of the multi-channel mechanism enables the Volterra series to capture dynamic features of time series without strictly satisfying time invariance.

## 5 REDUNDANCY-AWARE SPARSIFICATION IN DIVO

High-order polynomial modeling provides strong expressive capacity, but it inevitably introduces redundancy in both the input representation and parameter space. In Volterra-based models, many feature interactions are permutation-equivalent, and high-dimensional coefficient matrices contain internal correlations that can be removed without loss of modeling power. To address this issue, we propose a unified **redundancy-aware sparsification** strategy to retain only the most informative and independent components in the model.

Our design consists of two modules: (1) a **Fixed Masking** mechanism that eliminates structurally redundant features by exploiting the symmetry of monomials, and (2) a **Sparsified Low-Rank Decomposition** mechanism that combines low-rank factorization with learnable pruning in the decomposed domain to capture sparse structure in a frequency-aware basis.

## 5.1 FIXED MASKING OF VOLTERRA FEATURES

The Kronecker-powered Volterra feature $X^{\otimes k}$ number of semantically equivalent terms due to permutation invariance. Specifically, any reordering of the same input elements yields the same interaction:

$$x_1 x_2 \cdots x_k = x_{\pi(1)} x_{\pi(2)} \cdots x_{\pi(k)}, \quad \forall \pi \in S_k \tag{11}$$

where $S_k$ denotes the permutation group of $k$ elements.

To eliminate these redundant terms, we define a **fixed mask** tensor $\mathbf{M}_f^k \in \{0, 1\}$ that retains only monomials with non-decreasing index order. Each element is defined as:

$$\left(\mathbf{M}_f^k\right)_{i_1, i_2, \ldots, i_k} = \begin{cases} 1, & \text{if } i_1 \leq i_2 \leq \cdots \leq i_k \\ 0, & \text{otherwise} \end{cases} \tag{12}$$

This fixed mask is applied element-wise to the Volterra feature tensor $X^{\otimes k}$ before any parameter learning. It effectively removes symmetry-induced redundancy in the input space, reducing both storage and computation without sacrificing representational power.

## 5.2 REDUNDANCY-AWARE LOW-RANK DECOMPOSITION

To address redundancy in the parameter space, we propose a redundancy-aware low-rank decomposition strategy. We first factorize each Volterra coefficient matrix using a Tucker-style low-rank decomposition, and then apply learnable sparsification in the decomposed latent space.

Each coefficient matrix $A^n$ is decomposed into: a left projection matrix $U^n \in \mathbb{R}^{L^n \times R}$, a right projection matrix $V^n \in \mathbb{R}^{W^n \times R}$, and a core tensor $G^n \in \mathbb{R}^{R \times R \times R}$, where $R$ is the decomposition rank.

This decomposition maps the original high-dimensional coefficient space into a structured latent domain, where $G^n$ captures the core interaction patterns among compressed input dimensions. Volterra kernels often exhibit structured sparsity in the decomposed latent space, where only a subset of interaction patterns contribute significantly to the system's nonlinear behavior. We interpret this latent domain as a frequency-like space, where global and redundant structures can be more easily separated and sparsified.

To sparsify this latent representation, we introduce a learnable binary mask tensor $\mathbf{M}_l^n \in \{0, 1\}^{R \times R \times R}$ applied to the core tensor $G^n$. The mask is constructed using a historical score tensor $P_h(t)$ that tracks the average magnitude of each latent parameter across training steps:

$$P_h(t) = \frac{1}{t-1} \sum_{k=0}^{t-1} |P(k)| \tag{13}$$

We then apply a threshold-based sparsification rule:

$$\left(\mathbf{M}_l^n\right)_{i,j,k} = \begin{cases} 1, & \text{if } P_h(t)_{i,j,k} > \tau \\ 0, & \text{otherwise} \end{cases} \tag{14}$$

where $\tau$ is a sparsity threshold. The resulting $\mathbf{M}_l^n$ removes redundant or low-contribution components from the latent interaction space.

Given an input Volterra feature tensor $\mathcal{X}^{\otimes n} \in \mathbb{R}^{L^n \times W^n}$, the final masked output $\mathcal{O}^n \in \mathbb{R}^{R \times W^n}$ is computed as:

$$O^n = \left(\left(\mathbf{M}_l^n \odot G^n\right) \times_1 U^{n\top} \left(\mathbf{M}_f^n \odot \mathcal{X}^{\otimes n}\right)\right) \times_2 V^n \tag{15}$$

where $\mathbf{M}_f^n$ is the fixed feature-level mask described in Section 5.1. This formulation reflects a fully compressed and sparsified computation path, with redundancy reduced both in the feature and parameter domains.

# 6 EXPERIMENTS

## 6.1 EXPERIMENTAL SETUP

**Datasets, baselines and metrics**. We evaluate our model using three classic chaotic system synthetic datasets and three real-world datasets. The chaotic system synthetic datasets include Double Pendulum(DP)Yu & Bi (1998), Lorenz63(L63)Lorenz (1963), and N-bodyAarseth (2003). The real-world datasets include ETT, Weather(Wea.), Exchange(Ex.)Ma et al. (2024), BenzeneConcentration(BC.)Duarte-Davidson et al. (2001) and AppliancesEnergy(AE.)Candanedo et al. (2017). Details in Appendix F.1. We benchmark our proposed model against several baseline methods, including the classical statistical model ARIMA, LSTM, GRU, FEDformer, Autoformer, Informer, TSMixer, NeuralODE, KAN, WPMixer, TimeXer, ModernTCN, DeformTime, Crossformer, Dlinear, SparseTSF, Cyclenet, and FITS. For more detailed information about the datasets and baselines, please refer to Appendix F.2.

**Experimental Settings**. For each experiment, we ran 5 trials and calculated the mean and standard deviation. For more detailed experimental settings, please refer to Appendix F.3.

## 6.2 TIME SERIES FORECASTING

We applied DiVo to forecasting tasks on both chaotic system synthetic data and real-world time series data. For synthetic datasets, we used an input length of 48 and output length of 12, 24, 36, 48, while for real-world datasets, we used an input length of 96 and output length of 96, 192, 336, 720. The main MSE experimental results are shown in Table 1, and the MAE experimental results and experimental error can be found in Appendix G.4.

**Results** DiVo yields lower MSE than competing models in most cases across both synthetic and real-world benchmarks. In chaotic systems, it excels at capturing complex dynamics, and on practical datasets it often ranks first. The average lead in MSE index is 11.1% which highlights DiVo's strength in learning nonlinear dependencies and temporal structures. Appendix G.2 further shows that DiVo achieves a favorable balance between accuracy, parameter count, and inference speed. A long-horizon generalization test (Appendix G.5) further confirms its ability to predict extended dynamic behavior.

## 6.3 ABLATION ON THE NUMBER OF CHANNELS

Furthermore, we investigated the impact of the number of channels in the Multi-Channel Linear Transformation Layer module on model performance. We conducted experiments on the ETTh1 datasets, where the number of channels was selected from 1 to 30. Additionally, we evaluated the effectiveness of the multi-channel mechanism across different prediction horizons to demonstrate its robustness across varying temporal scales. The experimental results can be found in Appendix G.3 figure 6.

**Results** We can observe as the number of Channels increases from 1 to 30, the model's prediction error decreases by an average of 59.4%. The increase in Channel number significantly enhances the model's ability to capture complex dependencies. Additionally, our analysis across different prediction horizons confirms that the multi-channel mechanism maintains consistent effectiveness regardless of the forecasting length, demonstrating its robustness in handling both short-term and long-term prediction tasks.

## 6.4 IMPACT OF MASKING AND LOW-RANK MODULES

To evaluate the contribution of our optimization modules, we conduct a controlled study on the masking and low-rank decomposition components. We compare six model variants: the full model, a variant without both masking modules (w/o Masks), variants without the fixed mask (w/o Fixed) and learnable mask (w/o Learnable), a variant without low-rank decomposition (w/o LowRank). This setup allows us to isolate and quantify the effect of each component on model performance. We conducted experiments on the ETTh1 dataset, with prediction performance results shown in Table 2. Additionally, we compared the parameter counts and running times of these models, as shown in Appendix G.3 figure 6.

Table 1: Extended forecasting results MSE across different models and datasets.

| Dataset | | KAN | Neural ODE | ARIMA | GRU | Informer | TSMixer | WPMixer | ModernTCN | DeformTime | Dlinear | SparseTSF | FITS | DiVo |
|---|---|---|---|---|---|---|---|---|---|---|---|---|---|---|
| *Chaotic synthetic datasets* | | | | | | | | | | | | | | |
| DP | 12 | 0.118 | 0.014 | 1.99e-04 | 0.030 | 0.123 | 2.02e-04 | 1.29e-05 | 2.15e-04 | 3.42e-04 | 1.87e-04 | 2.91e-04 | 2.33e-04 | **1.14e-05** |
| | 24 | 0.116 | 0.035 | 7.36e-04 | 0.040 | 0.127 | 2.39e-04 | 2.62e-05 | 3.45e-04 | 5.78e-04 | 2.98e-04 | 4.12e-04 | 3.89e-04 | **1.62e-05** |
| | 36 | 0.097 | 0.032 | 1.67e-03 | 0.043 | 0.129 | 3.39e-04 | 6.18e-05 | 4.78e-04 | 7.91e-04 | 4.12e-04 | 5.89e-04 | 5.56e-04 | **3.39e-05** |
| | 48 | 0.098 | 0.049 | 2.90e-03 | 0.046 | 0.126 | 3.21e-04 | 8.68e-05 | 6.23e-04 | 1.05e-03 | 5.67e-04 | 7.89e-04 | 7.45e-04 | **7.62e-05** |
| L63 | 12 | 0.058 | 0.016 | 0.977 | 1.23e-03 | 0.264 | 0.085 | 0.563 | 1.45e-03 | 2.78e-03 | 0.089 | 0.125 | 0.134 | **3.05e-04** |
| | 24 | 0.123 | 0.028 | 1.209 | 4.06e-03 | 0.308 | 0.272 | 0.904 | 3.87e-03 | 5.45e-03 | 0.167 | 0.201 | 0.198 | **9.26e-04** |
| | 36 | 0.187 | 0.130 | 1.247 | 0.017 | 0.516 | 0.423 | 1.045 | 0.098 | 0.145 | 0.278 | 0.334 | 0.312 | **9.10e-03** |
| | 48 | 0.164 | 0.223 | 1.255 | 0.060 | 0.508 | 0.369 | 1.071 | 0.234 | 0.312 | 0.445 | 0.498 | 0.478 | **0.055** |
| Nbody | 12 | 0.266 | 0.679 | 0.439 | 0.247 | 0.521 | 0.827 | 0.456 | 0.298 | 0.378 | 0.389 | 0.423 | 0.356 | **0.205** |
| | 24 | 0.278 | 1.169 | 0.853 | 0.279 | 0.427 | 1.229 | 0.864 | 0.456 | 0.567 | 0.578 | 0.612 | 0.534 | **0.235** |
| | 36 | 0.260 | 1.275 | 1.274 | 0.335 | 0.457 | 1.574 | 1.298 | 0.678 | 0.789 | 0.734 | 0.798 | 0.712 | **0.240** |
| | 48 | 0.272 | 1.270 | 1.705 | 0.354 | 0.521 | 1.629 | 1.708 | 0.823 | 0.945 | 0.889 | 0.967 | 0.856 | **0.264** |
| *Real-world datasets* | | | | | | | | | | | | | | |
| ETT | 96 | 0.310 | 0.698 | 0.967 | 0.594 | 0.727 | 0.295 | 0.302 | 0.275 | 0.274 | 0.268 | **0.265** | 0.270 | 0.305 |
| | 192 | 0.351 | 0.835 | 0.982 | 0.668 | 0.789 | 0.347 | 0.347 | 0.348 | 0.327 | 0.303 | **0.302** | 0.345 | 0.369 |
| | 336 | 0.385 | 0.883 | 1.020 | 0.796 | 0.888 | 0.391 | 0.392 | 0.376 | 0.387 | 0.395 | 0.421 | 0.412 | **0.368** |
| | 720 | 0.460 | 0.894 | 1.056 | 0.877 | 0.903 | 0.467 | 0.485 | 0.512 | 0.534 | 0.523 | 0.541 | 0.552 | **0.455** |
| Wea. | 96 | 0.179 | 0.306 | 1.742 | 0.244 | 0.635 | 0.169 | 0.171 | 0.174 | 0.189 | 0.182 | 0.191 | 0.185 | **0.156** |
| | 192 | 0.221 | 0.370 | 1.742 | 0.323 | 0.700 | 0.220 | 0.217 | 0.232 | 0.248 | 0.241 | 0.252 | 0.245 | **0.204** |
| | 336 | 0.272 | 0.397 | 1.698 | 0.413 | 0.852 | 0.262 | 0.271 | 0.281 | 0.298 | 0.287 | 0.301 | 0.293 | **0.227** |
| | 720 | 0.334 | 0.478 | 1.725 | 0.460 | 0.613 | 0.324 | 0.345 | 0.341 | 0.367 | 0.351 | 0.371 | 0.359 | **0.301** |
| Ex. | 96 | 0.129 | 1.491 | 4.772 | 1.289 | 1.023 | 0.206 | **0.089** | 0.118 | 0.112 | 0.090 | 0.092 | 0.102 | **0.089** |
| | 192 | 0.334 | 1.980 | 4.657 | 1.173 | 1.308 | 0.371 | 0.184 | 0.178 | 0.201 | 0.189 | 0.205 | 0.194 | **0.166** |
| | 336 | 0.617 | 1.808 | 4.422 | 1.567 | 1.589 | 0.781 | 0.355 | 0.341 | 0.378 | 0.356 | 0.384 | 0.367 | **0.307** |
| | 720 | 0.724 | 2.509 | 4.825 | 2.391 | 2.915 | 1.480 | 0.917 | 0.534 | 0.612 | 0.567 | 0.598 | 0.589 | **0.469** |
| AE. | 96 | 0.699 | 1.810 | 2.662 | 1.958 | 1.659 | 0.517 | 0.437 | 0.456 | 0.489 | 0.471 | 0.495 | 0.482 | **0.402** |
| | 192 | 1.001 | 1.884 | 2.747 | 2.053 | 1.767 | 0.707 | 0.615 | 0.651 | 0.678 | 0.663 | 0.685 | 0.671 | **0.612** |
| | 336 | 1.452 | 2.188 | 2.965 | 2.230 | 1.889 | 0.901 | 0.874 | 0.723 | 0.789 | 0.745 | 0.798 | 0.767 | **0.651** |
| | 720 | 2.001 | 2.362 | 3.571 | 2.283 | 2.075 | 1.329 | 1.401 | 1.398 | 1.456 | 1.421 | 1.467 | 1.434 | **1.324** |
| BC. | 96 | 0.038 | 0.142 | 0.037 | 0.075 | 0.196 | 0.045 | 0.038 | 0.041 | 0.048 | 0.044 | 0.049 | 0.047 | **0.035** |
| | 192 | 0.074 | 0.220 | 0.073 | 0.158 | 0.244 | 0.085 | 0.075 | 0.078 | 0.087 | 0.081 | 0.089 | 0.084 | **0.070** |
| | 336 | 0.142 | 0.293 | 0.131 | 0.232 | 0.271 | 0.130 | 0.135 | 0.141 | 0.153 | 0.144 | 0.156 | 0.151 | **0.128** |
| | 720 | 0.310 | 0.595 | 0.311 | 0.471 | 0.416 | 0.336 | 0.343 | 0.332 | 0.364 | 0.341 | 0.369 | 0.353 | **0.307** |

Table 2: Ablation Study Results on ETTh1 dataset with different prediction lengths. The values represent the MSE of the prediction results.

| Dataset | | DiVo (Full) | w/o Masks | w/o Fixed | w/o Learnable | w/o LowRank |
|---|---|---|---|---|---|---|
| ETTh1 | 96 | **0.245** | 0.279 | 0.213 | 0.224 | 0.166 |
| | 192 | **0.233** | 0.302 | 0.257 | 0.267 | 0.206 |
| | 336 | **0.251** | 0.341 | 0.288 | 0.305 | 0.219 |
| | 720 | **0.289** | 0.412 | 0.428 | 0.464 | 0.317 |

**Results** Our results show that ablating either the fixed or learnable masking components or the low-rank (LoRa) decomposition module consistently worsens forecasting accuracy at all horizons. In Appendix G.1, we analysis the mask module using spectral analysis methods. From an efficiency standpoint, dropDiVog the mask modules yields a slight inference speed-up but actually increases the model's parameter count (since the pruning effect of the masks is lost), whereas removing the low-rank module dramatically increases both inference time and parameter count. In summary, these results underscore that both the masking and low-rank components are crucial for balancing accuracy and efficiency in forecasting models.

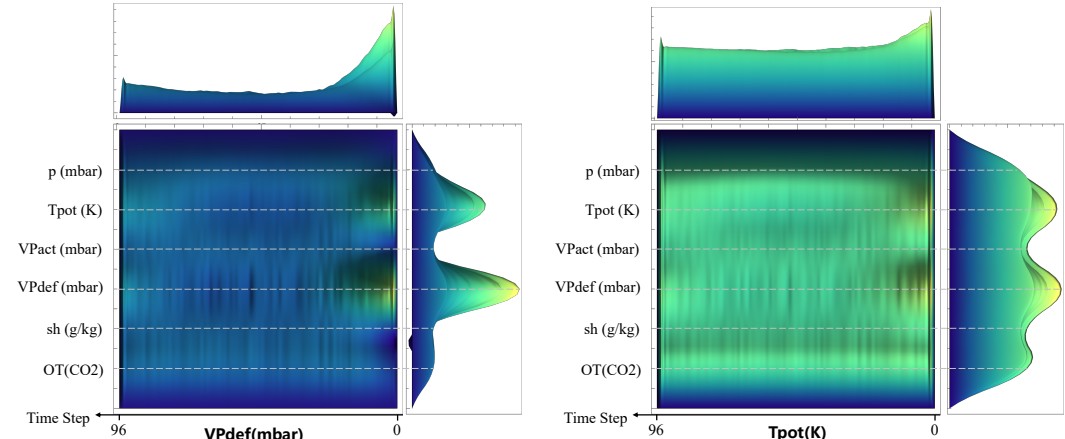

Figure 3: 3D heatmap of Weather dataset's learned discrete volterra coefficients matrix. The left figure shows the learned discrete volterra coefficients matrix of VPdef, while the right figure shows the learned discrete volterra coefficients matrix of Tpot. The upper projection is the projection of the parameters in time, and the right projection is the projection of the parameters in the input dimension.

### 6.5 INTERPRETABILITY EXPERIMENTS

To explore whether our model captures meaningful and interpretable dynamics from complex systems, we conduct interpretability experiments based on the learned parameters. We analyze the learned discrete volterra coefficients matrix across multiple datasets including Weather, ETT, and Exchange to investigate how different variables interact within these systems. Additionally, we validate the model's ability to recover underlying system dynamics by comparing the learned parameters with the known governing equations of the Lorenz63 chaotic system. By mathematically arranging the Discrete Volterra Coefficient Matrix, the discrete form of the system's differential equations can be derived. For systems with known mechanisms, another case study validated the interpretability of DiVo through a case study in the Appendix G.7, where we compared the system's actual differential equations with those learned through DiVo.

**Results** The learned discrete volterra coefficients matrix reveals some meaningful domain-specific patterns detailed in Appendix G.6. For Weather data, VPdef shows strong correlations with historical VPdef and Tpot, with shorter memory than Tpot, consistent with meteorological principles Zhou et al. (2014). In ETT, transformer oil temperature exhibits temporal dependencies with external loads, aligning with thermal dynamics IEE (2012). Exchange patterns show major currency pairs (USD, EUR, JPY) dominating interactions, matching actual market shares Bank for International Settlements (2013). These findings validate DiVo's ability to recover meaningful relationships inherent in the data. Results on the Lorenz63 in the Appendix G.7 verify the interpretability of DiVo for chaotic systems.

## 7 CONCLUSION

In this paper, we propose the Discrete Volterra Network (DiVo), a deep learning architecture that integrates the Volterra series with neural networks to explicitly capture nonlinear dynamics in time series. By reformulating prediction through discrete Volterra coefficient matrices, DiVo converts complex nonlinear modeling into structured learning of polynomial coefficients, improving both clarity and interpretability. To address practical issues, we introduce adaptive channel selection to relax time-invariance assumptions and a redundancy-aware sparsification strategy to control parameter growth. Experiments show that DiVo outperforms conventional deep learning methods in performance, efficiency, and interpretability. Its main limitation lies in sensitivity to polynomial order, which may require domain-specific tuning. Future work will extend DiVo to non-stationary and adaptive systems. Potential applications include scientific modeling, control, and domains such as climate science, neuroscience, and physics-based simulation, where explicit modeling of dynamic interactions is key to understanding and forecasting complex behaviors.

**Ethics Statement** All authors have read and agreed to abide by the ICLR Code of Ethics. We confirm that this submission complies with the ethical standards outlined by ICLR, including integrity in research, fairness in reviewing, and respectful participation in the conference.

**Reproducibility Statement** We have made efforts to ensure the reproducibility of our work. The main text describes the model architecture, training procedures, and evaluation protocols in detail (Sections 4 and Section 5). Additional implementation details, hyperparameter settings, and ablation studies are provided in the Appendix F.3. Dataset are described in Appendix F.1, and we provide fully reference to the baseline in Appendix F.2. An anonymous version of our code and scripts has been included in Appendix H to facilitate replication. Please note that while every effort has been made to ensure reproducibility, minor discrepancies may arise due to implementation details or hardware environments.

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

# A  THE USE OF LLM

In this work, large language models (LLMs) were used solely as general-purpose assistive tools for language polishing and literature search. They did not contribute to the research ideation, methodology design, or scientific content of the paper. LLMs have not been listed as authors.

# B  RELATED WORK

In this section, an overview of related work is provided, including deep time series models, and dynamical methods.

## B.1  DEEP TIME SERIES MODELS

Deep time series models for time series forecasting can be broadly categorized into four groups: autoregressive methods, fully connected neural networks, and transformer models and others.

**Autoregressive Models** Traditional statistical models such as the Autoregressive Integrated Moving Average (ARIMA) Ho & Xie (1998) have long been used for time series forecasting. More recently, Recurrent Neural Networks (RNNs) and their variants, such as Long Short-Term Memory (LSTM) Graves & Graves (2012) and Gated Recurrent Units (GRU) Dey & Salem (2017), have been employed to model sequential dependencies in time series data. These models utilize hidden states to encode past information and system evolution. However, their complex recursive structures and state transitions make them hard to interpret, limiting insight into the system's internal mechanisms.

**Fully Connected Neural Networks** Fully Connected Neural Networks (FCNNs) have demonstrated strong approximation capabilities for static nonlinear mathods Goodfellow et al. (2016), include TSMixer Chen et al. (2023), WPMixer Murad et al. (2025), TimeXer Wang et al. (2024). Linear models like DLinear Zeng et al. (2023) demonstrate effectiveness through simple decomposition strategies. Convolutional approaches like ModernTCN Luo & Wang (2024) leverage temporal convolutional networks for efficient sequence modeling. Attention-based models such as DeformTime Shu & Lampos (2024) introduce deformable attention mechanisms to adaptively capture temporal patterns. Parameter-efficient methods include SparseTSF Lin et al. (2025), which achieves strong performance with minimal parameters through sparse modeling, CycleNet Lin et al. (2024) that leverages cyclic patterns, and FITS Xu et al. (2023) which uses frequency domain interpolation for efficient forecasting. However, these models are inherently black-box in nature. Despite their empirical success, FCNNs provide limited insight into the causal structure of the system, making it difficult to interpret how specific features influence the predictions.

**Transformer Models** Transformer-based architectures, such as the Transformer Vaswani (2017), have gained popularity in time series forecasting due to their ability to capture long-range dependencies and parallelize computations. Models like Informer Zhou et al. (2021) Autoformer Wu et al. (2021) and FEDformer Zhou et al. (2022b) leverage self-attention mechanisms to model temporal correlations. Crossformer Zhang & Yan (2023) introduces cross-dimension dependency modeling to capture relationships between different variables in multivariate time series. However, these models often lack explicit interpretability and may struggle with complex nonlinear dynamics.

**Other** Some novel models, such as Neural ODEs Chen et al. (2018), combine neural networks with differential equations to model continuous-time dynamics, but struggle with long-term dependencies and may be computationally expensive due to adaptive step size adjustments. Other approaches, like KAN Liu et al. (2024), may face scalability issues and require extensive computational resources for training.

## B.2  DYNAMICAL METHODS

To approximate complex nonlinear behaviors without explicitly known equations, mathematical tools such as the Volterra series have been employed Zheng & Zafiriou (1996); Wray & Green (1994); Cheng et al. (2017); Borys (2018). The Volterra series models input-output relationships using polynomial functionals and captures memory effects through successive integrals. Some studies combined Volterra series with neural networks and proposed polynomial networksOh & Pedrycz

(2002) Chrysos et al. (2022). However, these methods suffers from exponential growth in complexity with increasing system order.

## C  PROOF OF THEOREM

The following proof establishes the discrete Volterra series representation for dynamical systems through Fredholm approximation. We begin with foundational concepts and progressively develop the matrix formulation.

**Fredholm Theorem Context** The Fredholm theorem for integral equations states that continuous operators of the form:

$$g(t) = \int_a^b K(t, \tau) f(\tau) d\tau \tag{16}$$

admit solutions when $K(t, \tau)$ is a compact operator. In our context, this guarantees that any functional non-linear interaction $f(x(\tau))$ acting on a temporal process $\{x(\tau)\}_{\tau=0}^t$ can be approximated by a linear combination of basis functions. Specifically, there exist coefficients $\{\alpha_i\}$ and orthogonal basis functions $\{\phi_i\}$ such that

$$f(x(\tau)) \approx h_0 + \sum_{i=1}^M \alpha_i \phi_i(x(\tau)) \tag{17}$$

provided that: (1) $f$ is continuous in $C([0, t])$, (2) the basis $\{\phi_i\}$ spans the function space, and (3) temporal correlations decay sufficiently fast. These conditions hold naturally for physical dynamical systems with fading memory.

**Dynamical System Framework** Consider a system governed by state evolution

$$x(t + \delta t) = \int_0^t f(x(\tau)) d\tau \tag{18}$$

where $f(\cdot)$ encodes nonlinear interactions. Applying Fredholm approximation, we decompose $f$ as

$$f(x(\tau)) = h_0 + \sum_{i=1}^M \alpha_i \left[ h_i + \sum_{n=1}^\infty p_n(x(\tau)) \right] \tag{19}$$

where $p_n(x(\tau))$ represents $n$-th order nonlinearities.

**Volterra Series Construction** Each $p_n(x(\tau))$ corresponds to a Volterra operator:

$$p_n(x(\tau)) = \int_{\mathbb{R}^n} h_n(\tau_1, \ldots, \tau_n) \prod_{i=1}^n x(t - \tau_i) d\tau_1 \cdots d\tau_n \tag{20}$$

where $h_n$ is the $n$-th order Volterra kernel. For implementation, we discretize the temporal domain with a truncation length $N$ and vectorize the state history as $X(t) = [x(t), x(t-1), ..., x(t-N+1)]^\top$. The discrete analog becomes

$$p_n(x_\tau) = \sum_{i_1, \ldots, i_n = 1}^N H_n(i_1, ..., i_n) \bigotimes_{k=1}^n X(t - i_k) \tag{21}$$

where $H_n \in \mathbb{R}^{N^n}$ is the discrete kernel tensor and $\bigotimes$ denotes Kronecker products.

**Tensor Representation** Equation 21 admits a compact representation through tensor contraction:

$$p_n(x_\tau) = \langle H_n, X^{\otimes n} \rangle_F \tag{22}$$

where $\langle \cdot, \cdot \rangle_F$ is the Hadamard product and $X^{\otimes n}$ is the Kronecker product $n$ times. Truncating at order $k$ yields the approximation

$$\phi_i(x_\tau) \approx \sum_{j=0}^l \lambda_j \left( A_0 + \sum_{n=1}^k A^n X^{\otimes n} \right) \tag{23}$$

with $A^n$ containing kernel coefficients and $\lambda_j$ as learnable weights. The truncation order $k$ balances model complexity and data availability.

**Deep Learning Integration** The critical insight lies in reformulating non-linear dynamics through the discrete Volterra coefficient matrix (discrete volterra coefficients matrix) paradigm. By constructing the matrix family $\mathcal{A}_k = \{A_k^1, A_k^2, \cdots, A_k^l\}$ with built-in time invariance, the complete discrete volterra coefficients matrix-based prediction model unifies all components as:

$$f(t) = h_0 + \sum_{j=0}^{l} \lambda_j \left( A_j + \sum_{n=1}^{k} A_j^n \odot X^{\otimes n} \right) \tag{24}$$

where $k$ is Maximum nonlinear order, $l$ is the number of parallel channels, $\lambda_j$ are learnable channel weights, $A_j$ are channel constant bias terms, and $A_j^n$ are $n$-th order channel Volterra coefficient matrices. The Hadamard product $\odot$ denotes tensor contraction along time-delay dimensions.

# D ROBUSTNESS TO INPUT PERTURBATIONS

In this section, we prove that the discrete Volterra reparameterized Discrete Volterra Network (DiVo) defined by Equation equation 6 is Lipschitz continuous with respect to its input window Lu et al. (2019), and hence stable under small perturbations. Let

$$X \in \mathbb{R}^{N \times d}, \qquad \|X\|_\infty = \max_{1 \leq i \leq N, \, 1 \leq \alpha \leq d} \left| X_{i,\alpha} \right|,$$

be the input time–series window of length $N$ and dimension $d$. We denote its $n$-th order Volterra feature by the Kronecker power $X^{\otimes n} \in \mathbb{R}^{d^n}$ and collect the learnable parameters as scalars $h_0$, $\{\lambda_j\}_{j=1}^{l}$, and vectors $\{A_j\}_{j=1}^{l}$, $\{A_j^n \in \mathbb{R}^{d^n}\}_{j=1,n=1}^{l,k}$. Then the network output is

$$f(X) = h_0 + \sum_{j=1}^{l} \lambda_j \left( A_j + \sum_{n=1}^{k} A_j^n \odot X^{\otimes n} \right) \in \mathbb{R}. \tag{25}$$

Consider two input windows $X, \widetilde{X}$. Since the biases $A_j$ do not depend on $X$, they contribute only to the constant term and do not affect stability. For each channel $j$ and order $n$, define the difference

$$\Delta_j^n = A_j^n \odot \left( X^{\otimes n} - \widetilde{X}^{\otimes n} \right).$$

By the properties of the Hadamard product,

$$\|\Delta_j^n\|_\infty = \max_{1 \leq I \leq d^n} \left| A_{j,I}^n \right| \left| \left( X^{\otimes n} - \widetilde{X}^{\otimes n} \right)_I \right| \leq \|A_j^n\|_\infty \left\| X^{\otimes n} - \widetilde{X}^{\otimes n} \right\|_\infty,$$

where $\|A_j^n\|_\infty = \max_I |A_{j,I}^n|$. Moreover, the difference of Kronecker powers can be bounded via the identity

$$a^n - b^n = (a - b) \sum_{i=0}^{n-1} a^i b^{n-1-i},$$

applied elementwise to vectors, yielding

$$\left\| X^{\otimes n} - \widetilde{X}^{\otimes n} \right\|_\infty \leq n \, M^{n-1} \|X - \widetilde{X}\|_\infty, \quad M = \max\{\|X\|_\infty, \|\widetilde{X}\|_\infty\}.$$

Combining these bounds, we obtain

$$\left| f(X) - f(\widetilde{X}) \right| = \left| \sum_{j=1}^{l} \lambda_j \sum_{n=1}^{k} A_j^n \odot \left( X^{\otimes n} - \widetilde{X}^{\otimes n} \right) \right|$$

$$\leq \sum_{j=1}^{l} |\lambda_j| \sum_{n=1}^{k} \|\Delta_j^n\|_\infty$$

$$\leq \sum_{j=1}^{l} |\lambda_j| \sum_{n=1}^{k} \|A_j^n\|_\infty \, n \, M^{n-1} \|X - \widetilde{X}\|_\infty.$$

Defining the constant

$$L \;=\; \sum_{j=1}^{l} |\lambda_j| \sum_{n=1}^{k} n \, \|A_j^n\|_\infty \, M^{n-1} < \infty,$$

we conclude

$$\left| f(X) - f(\widetilde{X}) \right| \;\le\; L \, \|X - \widetilde{X}\|_\infty,$$

which establishes that the DiVo model is Lipschitz continuous and therefore stable under small input perturbations.

## E  DiVo Model Interpretability Analysis

This appendix provides a comprehensive analysis of DiVo model's interpretability framework, encompassing both theoretical foundations and empirical validations across multiple domains. We demonstrate how DiVo's structured design enables direct extraction of physically meaningful insights from complex time series dynamics.

### E.1  Theoretical Foundation of DiVo Interpretability

DiVo's interpretability operates across three fundamental dimensions:

- **Individual Feature Impact**: Quantifying how single input features influence specific output dimensions
- **Temporal Dependencies**: Analyzing how feature values at different time points affect current predictions
- **Multi-feature Combined Effects**: Understanding interactions between multiple input features

The interpretability stems from the explicit physical meaning of each parameter in DiVo's Volterra polynomial integral kernel expansion:

$$f(t) = h_0 + \sum_{j=1}^{l} \lambda_j \left( A_j + \sum_{n=1}^{k} A_j^n \odot X^{\otimes n} \right) \tag{26}$$

where each component has direct interpretable meaning:

- $h_0$: System baseline output
- $A_j$: $j$-th channel bias term
- $A_j^n[i_1, i_2, \ldots, i_n]$: $n$-th order joint effect weight for features $(i_1, i_2, \ldots, i_n)$
- $\lambda_j$: Global weight for $j$-th channel

Each weight $A_j^n[i_1, i_2, \ldots, i_n]$ directly quantifies joint influence strength: positive values indicate enhancement, negative indicate suppression, with magnitude reflecting importance.

### E.2  Feature Dimension Projection

For multi-dimensional output, DiVo's prediction for the $d$-th dimension is:

$$y_d(t) = h_{0,d} + \sum_{j=1}^{l} \lambda_{j,d} \left( A_{j,d} + \sum_{n=1}^{k} A_{j,d}^n \odot X^{\otimes n} \right) \tag{27}$$

The total importance of feature $i$ for output dimension $d$ is calculated as:

$$\mathcal{I}_i = \sum_{j=1}^{l} |\lambda_{j,d} A_{j,d}^1[i]| + \sum_{n=2}^{K} \sum_{j=1}^{l} \sum_{\mathbf{k} \ni i} |\lambda_{j,d} A_{j,d}^n[\mathbf{k}]| \tag{28}$$

where $\mathbf{k} \ni i$ represents all $n$-element combinations containing feature $i$.

### E.3 Temporal Dimension Projection

DiVo's temporal dependencies are reflected through temporal dimension projection. The weight for feature $i$ at time $\tau$ is given by $A^1_{j,d}[\text{idx}(i,\tau)]$ where $\text{idx}(i,\tau) = i + d \times (\tau - 1)$.

The temporal pattern for feature $i$ is characterized by:

$$\mathcal{T}_i(\tau) = \{A^1_{j,d}[\text{idx}(i,1)], A^1_{j,d}[\text{idx}(i,2)], \ldots, A^1_{j,d}[\text{idx}(i,N)]\} \tag{29}$$

Memory features often exhibit exponential decay: $|A^1_{j,d}[\text{idx}(i,\tau)]| \propto e^{-\tau/\tau_0}$, periodic patterns, and long-term dependencies.

#### E.3.1 Multi-dimensional Joint Analysis

Unlike traditional neural networks where nonlinear activations obscure weight meanings, DiVo's linear structure enables direct analysis of multi-feature interactions:

- **Multi-feature Effects**: $\mathcal{M}_{i_1,\ldots,i_n} = A^n_{j,d}[i_1, i_2, \ldots, i_n]$
- **Feature-Time Effects**: $\mathcal{F}_{i,j}(\tau_1, \tau_2) = A^2_{j,d}[\text{idx}(i,\tau_1), \text{idx}(j,\tau_2)]$
- **Spatiotemporal Patterns**: $\mathcal{S} = \{A^n_{j,d}[\text{idx}(i_1,\tau_1), \ldots, \text{idx}(i_n,\tau_n)]\}$

This structured interpretability is achieved through explicit weight decomposition, enabling direct analysis without requiring post-hoc interpretation methods.

# F Details of Datasets, Baselines, and Experimental Settings

## F.1 Datasets

This paper uses five benchmark datasets to evaluate the performance of DiVo and three typical chaotic systems to verify the model's effectiveness, as shown in Figure 4. Table 3 provides an overview of these datasets.

**ETT** Ma et al. (2024) The ETT Dataset contains data from two power transformers (from two stations), with each data point comprising 8-dimensional features, including the recorded date of the data point, the predicted value "oil temperature," and six different types of external load values.

**Weather** Ma et al. (2024) The Weather Dataset records the weather conditions every 10 minutes throughout the year 2020, including 21 meteorological indicators such as temperature and humidity.

**Exchange** Ma et al. (2024) The Exchange Dataset collects daily exchange rates for Australia, the United Kingdom, Canada, Switzerland, China, Japan, New Zealand, and Singapore from 1990 to 2016.

**BenzeneConcentration** Arenas-García & Orieus (2016)This dataset contains one year of hourly multivariate time series data from air quality sensors, aiming to predict benzene concentration.

**AppliancesEnergy** Candanedo et al. (2017) This dataset contains 10-minute interval measurements of indoor conditions, energy usage, and weather data over 4.5 months, aimed at predicting appliances energy consumption in a smart home environment.

**Double Pendulum** Yu & Bi (1998) Double Pendulum The Double Pendulum Chaotic Dataset includes high-speed camera footage capturing the chaotic motion of a double pendulum, along with the pendulum's position data obtained through pattern matching on the markers.Shown in figure 4 left.

**Lorenz 63** Lorenz (1963) Lorenz63 A typical chaotic model. The experimental data used here were automatically generated by the method provided by HoGRC.Shown in figure 4 right.

**N-body** Aarseth (2003) The N-body Dataset simulates gravitational interactions among celestial bodies using Newtonian dynamics, generated via the REBOUND frameworkRein & Liu (2012). It records 3D trajectories and velocities of multiple bodies, exhibiting chaotic behavior and long-range nonlinear interactions, serving as a benchmark for testing spatiotemporal dependency modeling in complex physical systems.

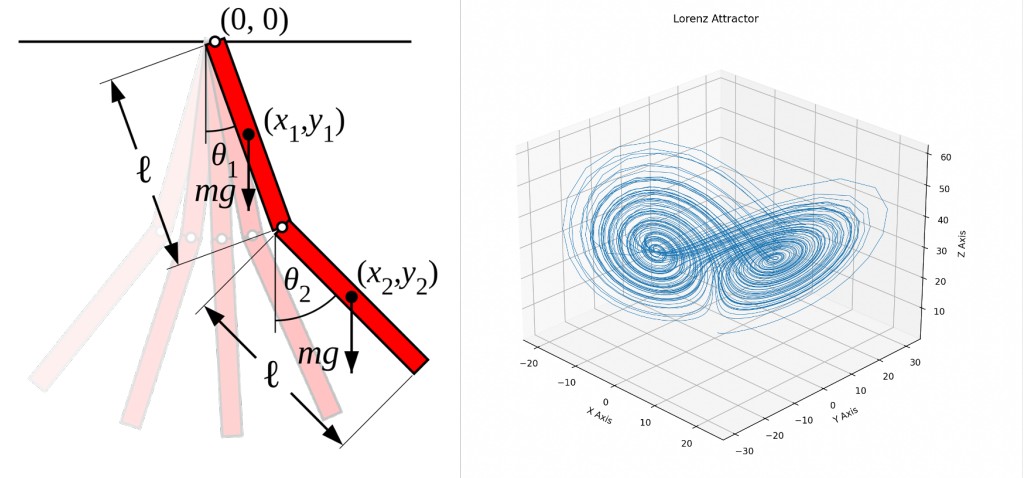

Figure 4: The left side of the figure shows a schematic of the ideal double pendulum system, while the right side visualizes the trajectory of the Lorenz63 system.

Table 3: Statistics of datasets for experiments.

| Datasets | Features | Timesteps |
|---|---|---|
| ETTh1 | 7 | 17420 |
| ETTh2 | 7 | 17420 |
| ETTm1 | 7 | 69680 |
| ETTm2 | 7 | 69680 |
| Weather | 21 | 52696 |
| Exchange | 8 | 7588 |
| AppliancesEnergy | 24 | 19728 |
| BenzeneConcentration | 8 | 1048575 |
| DoublePendulum | 2 | 15000 |
| Lorenz63 | 3 | 20000 |
| Nbody | 9 | 100000 |

## F.2 BASELINES

This paper selects the following state-of-the-art methods as baselines for comparison: **TSMixer**, **FEDformer**, **Autoformer**, **Informer**, **ARIMA**, **LSTM**, **GRU**, **Neural ODE**, **KAN**, **WPMixer** and **TimeXer** . Below we provide concise descriptions of these approaches:

**TSMixer**Chen et al. (2023): A fully MLP-based architecture that captures temporal patterns and cross-variable dependencies through alternating time- and feature-dimension MLP layers. Its core design combines fixed-weight characteristics of linear models across time steps with dynamic integration of cross-variable information, demonstrating strong performance in long-term forecasting tasks.

**FEDformer**Zhou et al. (2022a): A frequency-enhanced Transformer with seasonal-trend decomposition, reducing computational complexity through Fourier transform-based global dependency modeling ($\mathcal{O}(L \log L)$). Features two variants: Fourier-enhanced (FEDformer-F) and wavelet-enhanced (FEDformer-W).

**Autoformer** Wu et al. (2021): Introduces deep decomposition architecture and auto-correlation mechanisms to replace conventional attention. Achieves 38% improvement in energy and weather forecasting through progressive decomposition of latent variables and periodicity-based sub-sequence aggregation.

**Informer**Zhou et al. (2020): Transformer variant for long sequence prediction using ProbSparse attention to reduce complexity. Shows vulnerability to overfitting in scenarios with weak cross-variable correlations, sometimes underperforming simple linear models.

**ARIMA**Zhang (2003): Classical AutoRegressive Integrated Moving Average model handling non-stationary series through differencing. Combines AutoRegressive (AR) and Moving Average (MA) components with parameter selection guided by ACF/PACF analysis.

**LSTM**Greff et al. (2017): Long Short-Term Memory network employing forget/input/output gates to regulate information flow in cell states. Excels at capturing long-range temporal dependencies for complex pattern modeling.

**GRU**Dey & Salem (2017): Gated Recurrent Unit as a simplified LSTM variant, using update/reset gates to balance parameter efficiency and long-term dependency capture. (Note: Description supplemented from external knowledge)

**Neural ODE**Chen et al. (2018): Continuous-time model based on ordinary differential equations, enabling precise simulation of dynamical systems through adaptive numerical solvers.

**KAN**Liu et al. (2024): Novel architecture based on Kolmogorov-Arnold theorem, replacing fixed MLP weights with learnable edge functions. Demonstrates superior interpretability and performance in scientific tasks like knot theory and Anderson localization.

**WPMixer**Murad et al. (2025): A novel MLP-based model for long-term time series forecasting that introduces multi-resolution mixing to efficiently capture temporal patterns across different scales.

**TimeXer**Wang et al. (2024):TimeXer enhances Transformers for multivariate time series forecasting by effectively integrating exogenous variables through a novel cross-variable attention mechanism, multi-scale temporal encoding, and adaptive fusion modules.

**ModernTCN**Luo & Wang (2024): A modernized pure convolutional architecture that extends traditional TCN with enhanced receptive fields for efficient time series analysis.

**DeformTime**Shu & Lampos (2024): Employs deformable attention blocks to adaptively capture variable dependencies in multivariate time series, achieving 10% MAE reduction on average.

**Crossformer**Zhang & Yan (2023): A Transformer variant designed for multivariate time series forecasting with cross-dimensional attention mechanisms to model complex inter-variable relationships.

**DLinear**Zeng et al. (2023): Demonstrates that simple linear models can effectively extract strong temporal periodic dependencies through decomposition strategies, challenging complex architectures.

**SparseTSF**Lin et al. (2025): Achieves remarkable parameter efficiency with only 1k parameters by using cross-period sparse forecasting to decouple periodicity and trends.

Table 4: Expirements settings of time series datasets

| Dataset | Learning rate | Channels | Order |
|---|---|---|---|
| ETTm1 | 0.004 | 15 | 3 |
| ETTm2 | 0.004 | 15 | 3 |
| ETTh1 | 0.004 | 15 | 3 |
| ETTh2 | 0.004 | 15 | 3 |
| Weather | 0.005 | 25 | 2 |
| Exchange Rate | 0.001 | 25 | 3 |
| BenzeneConcentration | 0.001 | 25 | 3 |
| AppliancesEnergy | 0.001 | 50 | 3 |
| Lorenz63 | 0.001 | 3 | 2 |
| Double Pendulum | 0.001 | 10 | 3 |
| N-body | 0.001 | 13 | 3 |

**CycleNet**Lin et al. (2024): Introduces Residual Cycle Forecasting (RCF) to explicitly model inherent cyclical patterns using learnable recurrent cycles for enhanced long-term prediction.

**FITS**Xu et al. (2023): The first method to reduce LTSF model scale to 10k parameters through time-to-frequency domain transformation while maintaining excellent forecasting performance.

### F.3 EXPERIMENTAL SETTINGS

Table 4 shows the main parameters of the time series dataset. All datasets are divided into training, validation, and test sets in a ratio of 7:1:2. The training set is used to train the model, the validation set is used to select the best model, and the test set is used to evaluate the performance of the model. The optimizer is AdamW with weight decay of 0.001.

Expirements are conducted on a single NVIDIA RTX 4090 GPU with 24GB memory, with Intel i9-13900K CPU and 64GB RAM.

## G SUPPLEMENTARY RESULTS

### G.1 SPECTRAL ANALYSIS RESULTS

In this section, we provide a detailed analysis of the spectral properties of the learned discrete volterra coefficients matrix. The spectral analysis is performed on the discrete volterra coefficients matrix obtained from the trained DiVo model, which captures part dynamics of the Weather dataset (7 dimensions, 96 input length). The spectral properties are crucial for understanding the benifits of the mask method.

**Results** We visualized the parameters of the second-order discrete volterra coefficients matrix model trained on the Weather dataset using two-dimensional heat maps and conducted a spectral analysis by applying the two-dimensional Fast Fourier Transform to the model parameters, both with and without the mask module. The experimental results are presented in the figure 5. The upper-left panel displays the normalized parameter heat map of the model without the mask module, while the upper-right panel shows the corresponding frequency-domain spectrum. The lower-left panel illustrates the normalized parameter heat map with the mask module applied, and the lower-right panel presents its corresponding spectrum. By comparing the frequency-domain representations, we observe that the introduction of the mask module leads to a more dispersed spectral distribution. This suggests that the mask mechanism facilitates a more uniform learning of temporal evolution patterns across different time scales.

### G.2 MODEL EFFICIENCY EVALUATION

To assess the efficiency of our method, we conducted a comparative analysis against baseline models on the ETTh1 dataset, shown in table 5 . Our efficiency evaluation focused on several key metrics: the

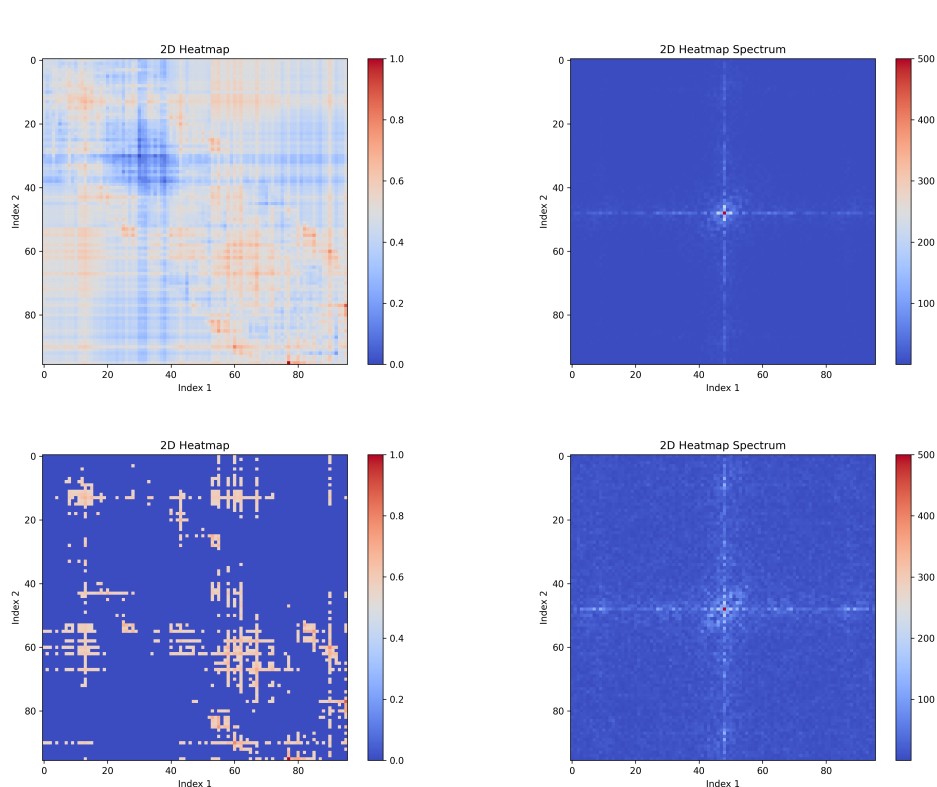

Figure 5: The spectral analysis results of the learned discrete volterra coefficients matrix: In the heatmap, the coordinates represent the time indices of the model parameters. For the second-order discrete Volterra coefficient matrix, each element in the matrix has two time indices, which can be visualized using a two-dimensional heatmap. Applying a Fourier transform to the heatmap yields a spectrum, reflecting the distribution of model parameters in the frequency domain. Comparing the heatmap and spectrum of the model parameters without the mask module, it is evident that introducing the mask module results in a more dispersed spectral distribution. This indicates that the mask mechanism helps the model learn temporal evolution patterns more uniformly across different time scales.

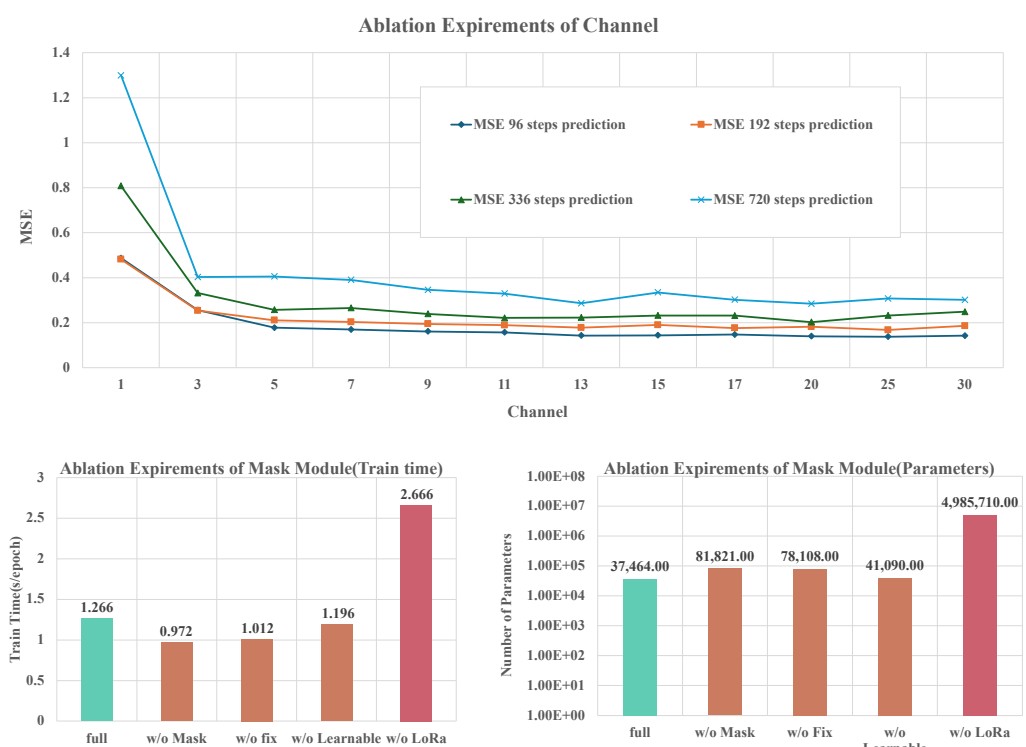

Figure 6: The figure(top) show the ablation experiment between prediction accuracy and the number of channels. The figure(bottom) show the relationship between training time, model parameters, mask, and LoRa.

average training time per epoch, the model's inference time, and the total number of model parameters. In our experimental setup, the batch size was uniformly set to 512, and the input sequence length was fixed at 96. Furthermore, to investigate the model's performance across varying prediction horizons, we configured the prediction sequence lengths to 96, 192, 384, 720. All experiments were conducted three times under identical conditions, and the average results are reported. Additionally, we documented the total number of parameters for each model.

**Results** The results of the efficiency evaluation are shown in table 5. According to the result, DiVo is a medium-parameter model (37.4K) with inference speed superior to mainstream Transformers (e.g., Informer) and comparable to lightweight models (e.g., TSMixer), though its parameter count exceeds traditional methods (e.g., ARIMA). It suits time-series tasks requiring a balance between efficiency and performance.

### G.3 SUPPLEMENTARY ABLATION RESULTS

This appendix provides supplementary illustrative figures for the experimental results presented in Section 6.4.

### G.4 SUPPLEMENTARY PREDICTION RESULTS

In this section, we provide the detailed prediction results of the DiVo model across different datasets. The results are presented in terms of Mean Absolute Error (MAE) in table 7, Mean Squared Error (MSE) in table 6. The experiments error also include the standard deviation (std) of the MSE and MAE, which is calculated over 5 independent runs. The standard deviation results are shown in table 8 and table 9. We present the MSE results in a visualized manner in Figure 7, where the performance

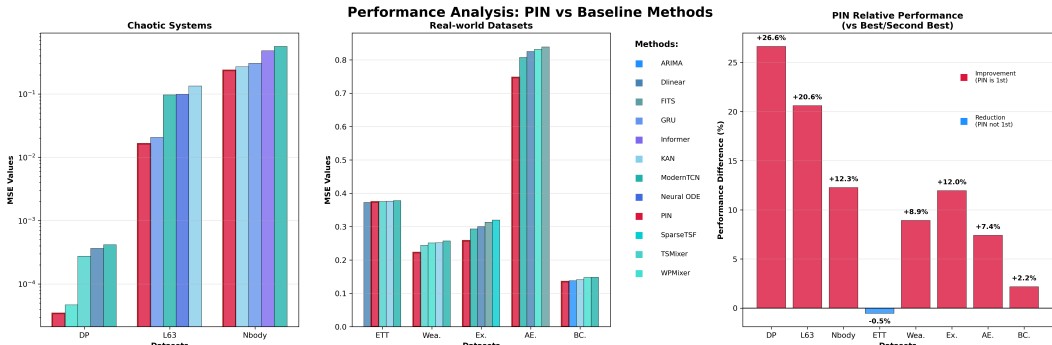

Figure 7: Left: Model ranking on chaotic system datasets. Middle: Model ranking on real-world datasets. Right: Performance margin of DiVo compared to other methods across all datasets (positive values indicate DiVo outperforms baselines, negative values indicate DiVo underperforms).

of each model can be intuitively compared. In addition, the figure also highlights the performance margins by which DiVo outperforms the competing methods on each dataset.

We also randomly selected 5 experimental results from the weather, exchange, ETT, double pendulum and Lorenz dataset to illustrate the model's performance. The results are shown in figure 8. The x-axis represents the time steps, while the y-axis represents the predicted values. The blue line indicates the predicted values, and the orange line indicates the ground truth values. The shaded area represents the standard deviation of the prediction results.

**results** The results of the experiments are shown in table 7, 8 and 9. The results show that the DiVo model outperforms all baseline models in terms of prediction accuracy, with the MSE increase 11.1%. The results also show that the DiVo model is robust to different prediction lengths, with a small increase in error as the prediction length increases.

Notice: 1.For the high-frequency DP dataset, KAN's learnable activations and Informer's sparse attention fail to capture potential bifurcations in chaotic systems, while Autoformer and FEDformer excel through decomposition and frequency domain approaches. 2.For the L63 dataset, being an autonomous system with relatively simple evolution rules, LSTM and GRU achieve unexpectedly good performance due to their architectural similarity to differential equation forms. 3.For the periodic ETT dataset, DiVo underperforms frequency based models like SparseTSF in short-term prediction but achieves state-of-the-art performance in long-term tasks by capturing underlying system dynamics.

The results of the random selection of experimental results are shown in figure 8. The results show that the DiVo model is able to capture the underlying patterns of the data and make accurate predictions. The shaded area represents the standard deviation of the prediction results, indicating that the DiVo model is robust to different input sequences.

### G.5 LONG-HORIZON GENERALIZATION EXPERIMENTS

To evaluate the model's ability to capture the intrinsic dynamics of complex systems and generalize across temporally distant scenarios, we designed a Long-Horizon Generalization Test. In this setting, the dataset was temporally partitioned into four segments: 30% for training, 40% discarded to create a significant temporal gap, 10% for validation, and the remaining 20% for testing. The inclusion of a 40% discarded segment between the training and testing periods introduces a substantial time horizon, which effectively suppresses low-order temporal correlations and poses a challenging extrapolation task.

This experimental configuration is designed to examine whether a model can truly learn the governing dynamics of the system, rather than overfitting to short-range statistical patterns. Successful generalization in this setting reflects the model's ability to extract stable, system-level principles that are invariant across different temporal regimes. The test was performed on the Lorenz63 datasets, with an input sequence length of 48 and an output sequence length of 12, 24, 36, 48.

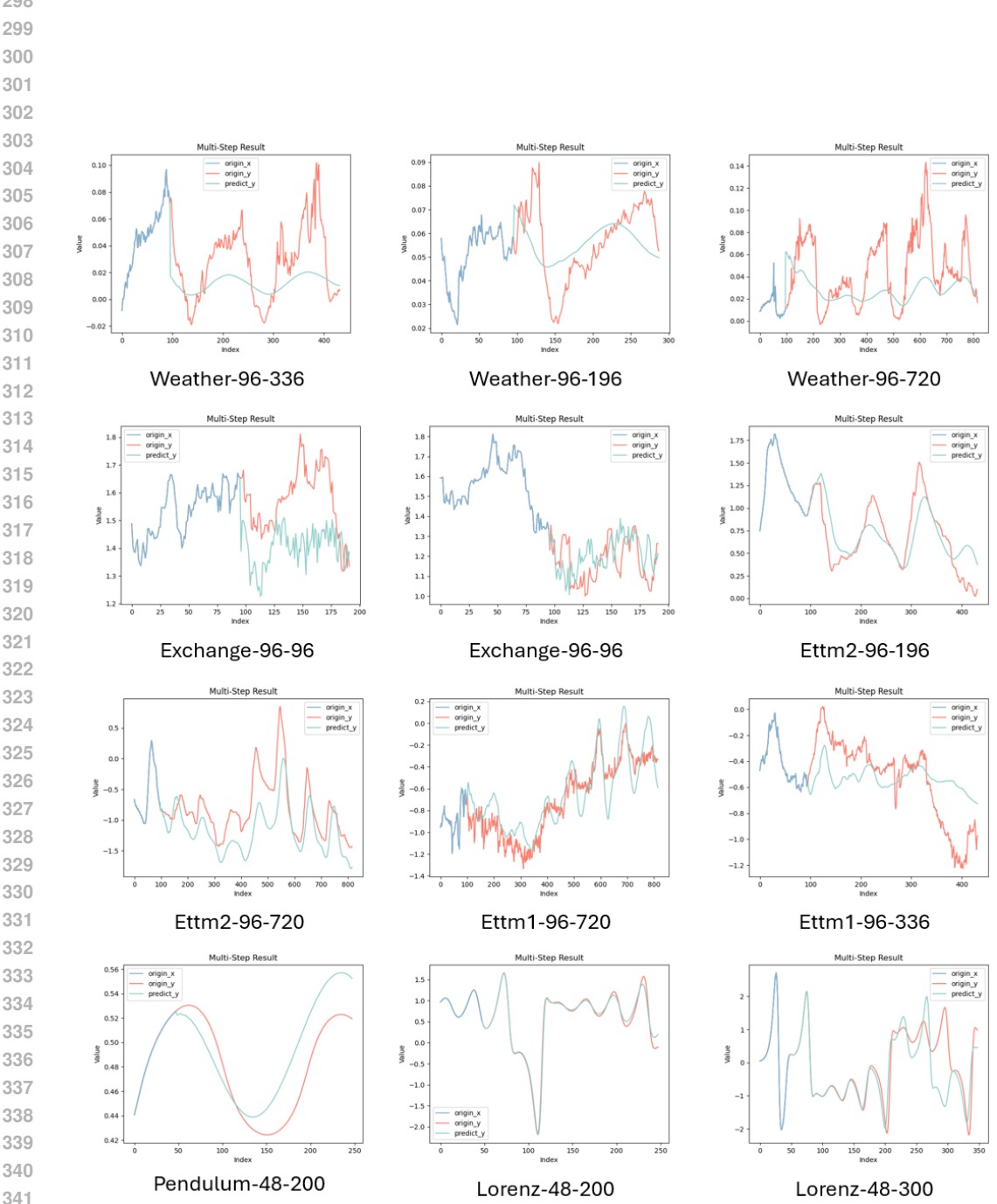

Figure 8: Randomly selected experimental results

**Results** The experimental results, summarized in Table 10, indicate that the proposed Discrete Volterra Network (DiVo) significantly outperforms baseline models under the long-horizon generalization setting. DiVo significantly outperforms all baseline models across all time steps, achieving 68.3%–86.8% MSE improvement and 58.4%–93.1% MAE improvement over the second-best model, demonstrating its superior capability in capturing system dynamics.

These results suggest that DiVo effectively captures the fundamental dynamical structure of the system, enabling it to extrapolate accurately over large temporal gaps. Unlike conventional sequence models that primarily rely on local temporal dependencies, DiVo's structure—grounded in explicit modeling of temporal integrals and nonlinear interactions—provides a more principled framework for representing long-term evolution behaviors.

Furthermore, qualitative inspection of the predicted trajectories shows that DiVo maintains phase alignment and amplitude consistency over extended horizons, while other models tend to diverge or smooth out high-frequency components. This further supports the conclusion that DiVo internalizes deeper structural regularities of the dynamical process.

### G.6 INTERPRETABILITY EXPERIMENTS

**ETT Dataset**

We validate DiVo's interpretability using the ETTm1 dataset, which contains electrical transformer temperature data with 7 variables:

- **Active Loads**: HUFL, MUFL, LUFL (representing effective power transmission at different load levels)
- **Reactive Loads**: HULL, MULL, LULL (representing magnetizing no-load losses and reactive power consumption)
- **Oil Temperature**: OT (Top-Oil Temperature, prediction target)

The experimental configuration uses input length 96 steps, prediction length 96 steps, with 15-minute sampling covering 2016-2018 data.

Applying Equation 28 to analyze each feature's importance for oil temperature prediction yields, the results are shown in Table 11.

The results reveal a clear active-reactive pairing pattern where active loads consistently outweigh corresponding reactive loads at each level, confirming copper loss ($I^2R$) dominance in transformer heating. The hierarchy H > M > L perfectly aligns with transformer thermal theory.

Using Equation 29, we analyze temporal memory patterns by fitting exponential decay models, The results are shown in Table 12.

The temporal patterns validate fundamental transformer physics: electrical responses are rapid ($\tau \approx 5-7$ steps), while thermal responses are slow ($\tau \approx 13$ steps). The oil temperature time constant of 3.2 hours aligns with typical ONAN transformer specifications (2-4 hours) as documented in IEEE C57.91-2011.

DiVo's second-order joint effects reveal key nonlinear interactions:

- **HUFL-HULL (0.087)**: Active-reactive power coupling, representing $P^2 + Q^2$ nonlinear heating
- **HUFL-OT (0.063)**: Load-temperature feedback coupling reflecting temperature-dependent coefficients
- **MUFL-MULL (0.028)**: Medium load joint heating demonstrating stratified interactions

The experiments on the ETTm1 dataset demonstrate that DiVo effectively uncovers patterns consistent with transformer physics across feature importance, temporal memory, and nonlinear interactions.

**Exchange Dataset** We further validate DiVo's interpretability using foreign exchange data with 8 major currency pairs predicting USD/EUR exchange rates. This analysis demonstrates DiVo's capability in complex economic systems. The results are shown in Table 13.

DiVo model weights show strong alignment with BIS 2013 market share data, achieving an average relative error of 7.6%. The model demonstrates excellent calibration for major currency pairs with higher deviations only for smaller market participants.

Critical joint effects discovered by DiVo include:

- **EUR-JPY (0.079)**: Safe-haven currency clustering during market stress
- **EUR-GBP (0.062)**: Regional integration effect consistent with optimal currency area theory
- **JPY-CHF (0.049)**: Flight-to-quality mechanism during uncertainty

demonstrating that DiVo is able to capture meaningful and interpretable relationships among variables in complex time series data.

**Results** The interpretability experiments on both the ETTm1 and Exchange datasets demonstrate that DiVo effectively uncovers patterns consistent with domain knowledge across feature importance, temporal memory, and nonlinear interactions. The results validate DiVo's capability to provide meaningful insights into complex systems, enhancing its value beyond mere predictive accuracy.

## G.7  INTERPRETING SYSTEM DYNAMICS

For the Lorenz63 system, its governing equations can be expressed in the following differential equation Form:

$$\begin{cases} \frac{dx}{dt} = \sigma(y - x) \\ \frac{dy}{dt} = x(\rho - z) - y \\ \frac{dz}{dt} = xy - \beta z \end{cases}$$

where $\sigma = 10$, $\rho = 28$, and $\beta = 8/3$. By rewriting the differential equations in a discrete (difference) form, we have:

$$\begin{cases} \frac{\Delta x}{\Delta t} = \frac{x_{n+1} - x_n}{\Delta t} = \sigma(y_n - x_n) \\\\ \frac{\Delta y}{\Delta t} = \frac{y_{n+1} - y_n}{\Delta t} = x_n(\rho - z_n) - y_n \\\\ \frac{\Delta z}{\Delta t} = \frac{z_{n+1} - z_n}{\Delta t} = x_n y_n - \beta z_n \end{cases}$$

where $\Delta t$ is the time step. By rearranging the equations, we can express them in a matrix form:

$$\begin{bmatrix} \frac{\Delta x}{\Delta t} \\\\ \frac{\Delta y}{\Delta t} \\\\ \frac{\Delta z}{\Delta t} \end{bmatrix} = \mathbf{C} \odot \mathbf{\Phi}(x, y, z),$$

where $\mathbf{C}$ is the coefficient matrix, and $\mathbf{\Phi}(x, y, z)$ is the basis function vector:

$$\mathbf{\Phi} = \begin{bmatrix} 1, \ x, \ y, \ z, \ xx, \ xy, \ xz, \ yx, \ yy, \ yz, \ zx, \ zy, \ zz \end{bmatrix}^\top$$

$$\mathbf{C} = \begin{bmatrix} 0, -10, 10, 0, 0, 0, 0, 0, 0, 0, 0, 0, 0 \\ 0, 28, -1, 0, 0, 0, 0, 0, 0, 0, 0, 0, -1 \\ 0, 0, 0, -8/3, 0, 1, 0, 0, 0, 0, 0, 0, 0 \end{bmatrix}.$$

$\mathbf{C}$ is the coefficient matrix, and $\mathbf{\Phi}(x, y, z)$ is the basis function vector. By multiplying the basis function vector with the coefficient matrix, we can obtain the discrete differential equations of the system. From trained DiVo, we can obtain the learned discrete Volterra coefficient matrix (discrete volterra coefficients matrix) $\mathbf{A_0}, \mathbf{A_1}, \mathbf{A_2}$, shown as follows:

$$\mathbf{A_0} = \begin{bmatrix} 2.1 \times 10^{-5} \\ -4.8 \times 10^{-5} \\ 7.3 \times 10^{-5} \end{bmatrix}.$$

$$\mathbf{A_1} = \begin{bmatrix} -10.0 & 10 & -1.6 \times 10^{-5} \\ 28.0 & -1.0 & 4.1 \times 10^{-5} \\ 6.3 \times 10^{-5} & 1.2 \times 10^{-5} & -2.7 \end{bmatrix}.$$

$$\mathbf{A_2} = \begin{bmatrix} O(10^{-6}) & O(10^{-6}) & O(10^{-5}) & \cdots & O(10^{-7}) \\ O(10^{-5}) & O(10^{-5}) & -1.0 & \cdots & O(10^{-6}) \\ O(10^{-5}) & 1.0 & O(10^{-6}) & \cdots & O(10^{-5}) \end{bmatrix}.$$

Compared to the original system, the learned discrete volterra coefficients matrix $\mathbf{A_0}$, $\mathbf{A_1}$, $\mathbf{A_2}$ approximated the original system's coefficient matrix $\mathbf{C}$, with the following results:

$$\mathbf{C} \approx \begin{bmatrix} \mathbf{A_0} & \mathbf{A_1} & \mathbf{A_2} \end{bmatrix} \quad (\varepsilon_{\max} = 7.3e - 5)$$

**Results** Within the margin of error ($\varepsilon < 10^{-4}$), the principal terms in the first-order and second-order discrete volterra coefficients matrix (those corresponding to $\sigma$, $\rho$, and $\beta$) are consistent, indicating that the neural network has successfully recovered the dynamical structure of the Lorenz63 system.

## H  CODE LINK

https://anonymous.4open.science/r/DIVO-E1FF/

Table 5: Extended Model Efficiency Analysis: This table systematically compares the computational efficiency of 19 time-series forecasting models across different prediction lengths (96/192/384/768), presenting three key metrics: training time (Tra.Time in seconds), inference time (Inf.Time in seconds), and parameter count (Para.).

| Metric | | Tra. Time(s) | Inf. Time(s) | Para. | Metric | | Tra. Time(s) | Inf. Time(s) | Para. |
|---|---|---|---|---|---|---|---|---|---|
| KAN | 96 | 1.303 | 0.629 | 1.18e+05 | Fedformer | 96 | 6.863 | 0.752 | 1.85e+07 |
| | 192 | 1.444 | 0.687 | 1.35e+05 | | 192 | 8.296 | 0.798 | 1.85e+07 |
| | 384 | 1.491 | 0.723 | 1.69e+05 | | 384 | 6.869 | 0.865 | 1.85e+07 |
| | 768 | 1.537 | 0.805 | 2.37e+05 | | 768 | 6.388 | 0.772 | 1.85e+07 |
| Neural ODE | 96 | 1.15e+01 | 1.188 | 4.58e+03 | TSMixer | 96 | 0.794 | 0.532 | 1.62e+04 |
| | 192 | 1.41e+01 | 1.477 | 4.58e+03 | | 192 | 0.839 | 0.412 | 2.55e+04 |
| | 384 | 2.32e+01 | 2.312 | 4.58e+03 | | 384 | 0.888 | 0.432 | 4.41e+04 |
| | 768 | 3.79e+01 | 3.139 | 4.58e+03 | | 768 | 1.100 | 0.692 | 8.14e+04 |
| ARIMA | 96 | 0.612 | 0.530 | 1.40e+01 | WPMixer | 96 | 1.476 | 0.712 | 2.26e+04 |
| | 192 | 0.702 | 0.398 | 1.40e+01 | | 192 | 1.520 | 0.702 | 3.19e+04 |
| | 384 | 0.769 | 0.413 | 1.40e+01 | | 384 | 1.621 | 0.693 | 5.06e+04 |
| | 768 | 0.895 | 0.458 | 1.40e+01 | | 768 | 1.729 | 0.671 | 1.62e+05 |
| LSTM | 96 | 4.051 | 0.685 | 3.90e+03 | TimeXer | 96 | 1.493 | 0.674 | 9.95e+04 |
| | 192 | 6.319 | 0.939 | 3.90e+03 | | 192 | 1.762 | 0.727 | 1.10e+05 |
| | 384 | 1.11e+01 | 1.536 | 3.90e+03 | | 384 | 1.677 | 0.745 | 1.32e+05 |
| | 768 | 1.88e+01 | 1.182 | 3.90e+03 | | 768 | 1.930 | 0.685 | 1.75e+05 |
| GRU | 96 | 3.752 | 0.772 | 2.95e+03 | ModernTCN | 96 | 1.823 | 0.594 | 8.64e+04 |
| | 192 | 7.078 | 1.041 | 2.95e+03 | | 192 | 2.147 | 0.678 | 9.23e+04 |
| | 384 | 9.995 | 0.908 | 2.95e+03 | | 384 | 2.385 | 0.742 | 1.12e+05 |
| | 768 | 1.79e+01 | 1.195 | 2.95e+03 | | 768 | 2.864 | 0.856 | 1.48e+05 |
| Autoformer | 96 | 2.201 | 0.673 | 1.32e+07 | DeformTime | 96 | 3.245 | 0.887 | 1.56e+05 |
| | 192 | 2.452 | 0.921 | 1.32e+07 | | 192 | 3.784 | 0.965 | 1.72e+05 |
| | 384 | 2.421 | 0.986 | 1.32e+07 | | 384 | 4.126 | 1.054 | 2.09e+05 |
| | 768 | 2.628 | 0.984 | 1.32e+07 | | 768 | 4.892 | 1.213 | 2.76e+05 |
| Informer | 96 | 2.085 | 0.786 | 1.20e+07 | Crossformer | 96 | 4.156 | 0.954 | 2.14e+05 |
| | 192 | 2.309 | 0.791 | 1.20e+07 | | 192 | 4.867 | 1.076 | 2.38e+05 |
| | 384 | 2.273 | 0.847 | 1.20e+07 | | 384 | 5.324 | 1.187 | 2.89e+05 |
| | 768 | 2.571 | 0.959 | 1.20e+07 | | 768 | 6.245 | 1.365 | 3.82e+05 |
| Dlinear | 96 | 0.523 | 0.412 | 4.32e+05 | SparseTSF | 96 | 2.194 | 0.734 | 6.45e+02 |
| | 192 | 0.612 | 0.456 | 4.54e+05 | | 192 | 2.587 | 0.823 | 7.32e+02 |
| | 384 | 0.734 | 0.523 | 4.72e+05 | | 384 | 2.896 | 0.891 | 8.24e+02 |
| | 768 | 0.896 | 0.634 | 4.88e+05 | | 768 | 3.412 | 1.026 | 9.31e+02 |
| Cyclenet | 96 | 3.567 | 0.823 | 1.34e+05 | FITS | 96 | 1.687 | 0.567 | 4.92e+04 |
| | 192 | 4.178 | 0.934 | 1.49e+05 | | 192 | 1.934 | 0.634 | 5.56e+04 |
| | 384 | 4.623 | 1.023 | 1.81e+05 | | 384 | 2.145 | 0.721 | 6.74e+04 |
| | 768 | 5.423 | 1.187 | 2.39e+05 | | 768 | 2.512 | 0.834 | 8.91e+04 |
| DiVo | 96 | 1.266 | 0.632 | 3.74e+04 | | | | | |
| | 192 | 2.302 | 0.823 | 3.74e+04 | | | | | |
| | 384 | 2.924 | 1.104 | 3.74e+04 | | | | | |
| | 768 | 3.426 | 1.207 | 3.74e+04 | | | | | |

Table 6: Extended forecasting results MSE across different models and datasets.

| Dataset | | KAN | Neural ODE | ARIMA | LSTM | GRU | Autoformer | Informer | Fedformer | TSMixer | WPMixer | TimeXer | ModernTCN | DeformTime | Crossformer | Dlinear | SparseTSF | Cyclenet | FITS | DiVo |
|---|---|---|---|---|---|---|---|---|---|---|---|---|---|---|---|---|---|---|---|---|
| | | | | | | | *Chaotic synthetic datasets* | | | | | | | | | | | | | |
| DP | 12 | 0.118 | 0.014 | 1.99e-04 | 0.051 | 0.030 | 3.41e-04 | 0.123 | 6.12e-04 | 2.02e-04 | 1.29e-05 | 8.17e-04 | 2.15e-04 | 3.42e-04 | 4.56e-04 | 1.87e-04 | 2.91e-04 | 3.78e-04 | 2.33e-04 | **1.14e-05** |
| | 24 | 0.116 | 0.035 | 7.36e-04 | 0.114 | 0.040 | 1.22e-03 | 0.127 | 1.59e-03 | 2.39e-04 | 2.62e-05 | 1.19e-03 | 3.45e-04 | 5.78e-04 | 7.23e-04 | 2.98e-04 | 4.12e-04 | 5.67e-04 | 3.89e-04 | **1.62e-05** |
| | 36 | 0.097 | 0.032 | 1.67e-03 | 0.164 | 0.043 | 2.42e-03 | 0.129 | 1.56e-03 | 3.39e-04 | 6.18e-05 | 1.91e-03 | 4.78e-04 | 7.91e-04 | 9.45e-04 | 4.12e-04 | 5.89e-04 | 7.34e-04 | 5.56e-04 | **3.39e-05** |
| | 48 | 0.098 | 0.049 | 2.90e-03 | 0.353 | 0.046 | 4.10e-03 | 0.126 | 4.45e-03 | 3.21e-04 | 8.68e-05 | 2.62e-03 | 6.23e-04 | 1.05e-03 | 1.23e-03 | 5.67e-04 | 7.89e-04 | 9.12e-04 | 7.45e-04 | **7.62e-05** |
| L63 | 12 | 0.058 | 0.016 | 0.977 | 9.31e-04 | 1.23e-03 | 0.312 | 0.264 | 0.235 | 0.085 | 0.563 | 0.733 | 0.014 | 0.028 | 0.156 | 0.089 | 0.125 | 0.198 | 0.134 | **3.05e-04** |
| | 24 | 0.123 | 0.028 | 1.209 | 2.90e-03 | 4.06e-03 | 0.507 | 0.308 | 0.405 | 0.272 | 0.904 | 0.977 | 0.039 | 0.054 | 0.234 | 0.167 | 0.201 | 0.289 | 0.198 | **9.26e-04** |
| | 36 | 0.187 | 0.130 | 1.247 | 0.011 | 0.017 | 0.632 | 0.516 | 0.552 | 0.423 | 1.045 | 1.086 | 0.098 | 0.145 | 0.389 | 0.278 | 0.334 | 0.456 | 0.312 | **9.10e-03** |
| | 48 | 0.164 | 0.223 | 1.255 | 0.690 | 0.060 | 0.727 | 0.508 | 0.644 | 0.369 | 1.071 | 1.064 | 0.234 | 0.312 | 0.567 | 0.445 | 0.498 | 0.634 | 0.478 | **0.055** |
| Nbody | 12 | 0.266 | 0.679 | 0.439 | 0.506 | 0.247 | 0.440 | 0.521 | 0.540 | 0.827 | 0.456 | 0.843 | 0.298 | 0.378 | 0.467 | 0.389 | 0.423 | 0.512 | 0.356 | **0.205** |
| | 24 | 0.278 | 1.169 | 0.853 | 0.553 | 0.279 | 0.490 | 0.427 | 1.168 | 1.229 | 0.864 | 1.252 | 0.456 | 0.567 | 0.634 | 0.578 | 0.612 | 0.689 | 0.534 | **0.235** |
| | 36 | 0.260 | 1.275 | 1.274 | 0.620 | 0.335 | 0.560 | 0.457 | 1.197 | 1.574 | 1.298 | 1.684 | 0.678 | 0.789 | 0.845 | 0.734 | 0.798 | 0.867 | 0.712 | **0.240** |
| | 48 | 0.272 | 1.270 | 1.705 | 0.657 | 0.354 | 1.097 | 0.521 | 1.566 | 1.629 | 1.708 | 2.127 | 0.823 | 0.945 | 1.023 | 0.889 | 0.967 | 1.034 | 0.856 | **0.264** |
| | | | | | | | *Real-world datasets* | | | | | | | | | | | | | |
| ETT | 96 | 0.310 | 0.698 | 0.967 | 0.834 | 0.594 | 0.296 | 0.727 | 0.279 | 0.295 | 0.302 | 0.348 | 0.275 | 0.274 | 0.283 | 0.268 | **0.265** | 0.289 | 0.270 | 0.305 |
| | 192 | 0.351 | 0.835 | 0.982 | 0.844 | 0.668 | 0.349 | 0.789 | 0.371 | 0.347 | 0.347 | 0.391 | 0.348 | 0.327 | 0.348 | 0.303 | **0.302** | 0.356 | 0.345 | 0.369 |
| | 336 | 0.385 | 0.883 | 1.020 | 0.884 | 0.796 | 0.406 | 0.888 | 0.433 | 0.391 | 0.392 | 0.431 | 0.376 | 0.387 | 0.389 | 0.395 | 0.421 | 0.398 | 0.412 | **0.368** |
| | 720 | 0.460 | 0.894 | 1.056 | 0.973 | 0.877 | 0.545 | 0.903 | 0.539 | 0.467 | 0.485 | 0.521 | 0.512 | 0.534 | 0.556 | 0.523 | 0.541 | 0.558 | 0.552 | **0.455** |
| Wea. | 96 | 0.179 | 0.306 | 1.742 | 0.413 | 0.244 | 0.160 | 0.635 | 0.169 | 0.169 | 0.171 | 0.194 | 0.174 | 0.189 | 0.203 | 0.182 | 0.191 | 0.207 | 0.185 | **0.156** |
| | 192 | 0.221 | 0.370 | 1.742 | 0.417 | 0.323 | 0.237 | 0.700 | 0.237 | 0.220 | 0.217 | 0.240 | 0.232 | 0.248 | 0.261 | 0.241 | 0.252 | 0.267 | 0.245 | **0.204** |
| | 336 | 0.272 | 0.397 | 1.698 | 0.453 | 0.413 | 0.293 | 0.852 | 0.293 | 0.262 | 0.271 | 0.289 | 0.281 | 0.298 | 0.312 | 0.287 | 0.301 | 0.318 | 0.293 | **0.227** |
| | 720 | 0.334 | 0.478 | 1.725 | 0.518 | 0.460 | 0.370 | 0.613 | 0.369 | 0.324 | 0.345 | 0.358 | 0.341 | 0.367 | 0.378 | 0.351 | 0.371 | 0.383 | 0.359 | **0.301** |
| Ex. | 96 | 0.129 | 1.491 | 4.772 | 1.930 | 1.289 | 0.103 | 1.023 | 0.106 | 0.206 | **0.089** | 0.101 | 0.118 | 0.112 | 0.095 | 0.090 | 0.092 | 0.119 | 0.102 | **0.089** |
| | 192 | 0.334 | 1.980 | 4.657 | 2.138 | 1.173 | 0.212 | 1.308 | 0.216 | 0.371 | 0.184 | 0.195 | 0.178 | 0.201 | 0.218 | 0.189 | 0.205 | 0.223 | 0.194 | **0.166** |
| | 336 | 0.617 | 1.808 | 4.422 | 2.278 | 1.567 | 0.400 | 1.589 | 0.413 | 0.781 | 0.355 | 0.357 | 0.341 | 0.378 | 0.398 | 0.356 | 0.384 | 0.405 | 0.367 | **0.307** |
| | 720 | 0.724 | 2.509 | 4.825 | 3.083 | 2.391 | 1.034 | 2.915 | 1.056 | 1.480 | 0.917 | 0.878 | 0.534 | 0.612 | 0.645 | 0.567 | 0.598 | 0.634 | 0.589 | **0.469** |
| AE. | 96 | 0.699 | 1.810 | 2.662 | 2.077 | 1.958 | 0.444 | 1.659 | 0.414 | 0.517 | 0.437 | 0.532 | 0.456 | 0.489 | 0.512 | 0.471 | 0.495 | 0.518 | 0.482 | **0.402** |
| | 192 | 1.001 | 1.884 | 2.747 | 2.246 | 2.053 | 0.672 | 1.767 | 0.624 | 0.707 | 0.615 | 0.696 | 0.651 | 0.678 | 0.698 | 0.663 | 0.685 | 0.704 | 0.671 | **0.612** |
| | 336 | 1.452 | 2.188 | 2.965 | 2.265 | 2.230 | 0.951 | 1.889 | 0.868 | 0.901 | 0.874 | 0.946 | 0.723 | 0.789 | 0.812 | 0.745 | 0.798 | 0.823 | 0.767 | **0.651** |
| | 720 | 2.001 | 2.362 | 3.571 | 2.703 | 2.283 | 1.473 | 2.075 | 1.500 | 1.329 | 1.401 | 1.455 | 1.398 | 1.456 | 1.489 | 1.421 | 1.467 | 1.501 | 1.434 | **1.324** |
| BC. | 96 | 0.038 | 0.142 | 0.037 | 0.371 | 0.075 | 0.043 | 0.196 | 0.036 | 0.045 | 0.038 | 0.046 | 0.041 | 0.048 | 0.052 | 0.044 | 0.049 | 0.053 | 0.047 | **0.035** |
| | 192 | 0.074 | 0.220 | 0.073 | 0.410 | 0.158 | 0.080 | 0.244 | 0.080 | 0.085 | 0.075 | 0.082 | 0.078 | 0.087 | 0.092 | 0.081 | 0.089 | 0.094 | 0.084 | **0.070** |
| | 336 | 0.142 | 0.293 | 0.131 | 0.625 | 0.232 | 0.149 | 0.271 | 0.149 | 0.130 | 0.135 | 0.146 | 0.141 | 0.153 | 0.162 | 0.144 | 0.156 | 0.165 | 0.151 | **0.128** |
| | 720 | 0.310 | 0.595 | 0.311 | 0.815 | 0.471 | 0.360 | 0.416 | 0.347 | 0.336 | 0.343 | 0.357 | 0.332 | 0.364 | 0.378 | 0.341 | 0.369 | 0.384 | 0.353 | **0.307** |

Table 7: Extended forecasting results MAE across different models and datasets.

| Dataset | | KAN | Neural ODE | ARIMA | LSTM | GRU | Autoformer | Informer | Fedformer | TSMixer | WPMixer | TimeXer | ModernTCN | DeformTime | Crossformer | Dlinear | SparseTSF | Cyclenet | FITS | DiVo |
|---|---|---|---|---|---|---|---|---|---|---|---|---|---|---|---|---|---|---|---|---|
| | | | | | | | *Chaotic synthetic datasets* | | | | | | | | | | | | | |
| DP | 12 | 0.130 | 0.059 | 8.95e-03 | 0.108 | 0.080 | 9.88e-03 | 0.253 | 9.22e-03 | 8.79e-03 | 2.28e-03 | 0.020 | 0.012 | 0.016 | 0.019 | 0.013 | 0.017 | 0.021 | 0.015 | **1.75e-03** |
| | 24 | 0.133 | 0.093 | 0.017 | 0.169 | 0.095 | 0.018 | 0.277 | 0.017 | 9.70e-03 | 3.24e-03 | 0.023 | 0.014 | 0.019 | 0.024 | 0.016 | 0.021 | 0.026 | 0.018 | **1.20e-03** |
| | 36 | 0.121 | 0.096 | 0.026 | 0.261 | 0.103 | 0.026 | 0.273 | 0.020 | 0.011 | 4.95e-03 | 0.029 | 0.017 | 0.023 | 0.028 | 0.019 | 0.025 | 0.031 | 0.022 | **4.82e-03** |
| | 48 | 0.122 | 0.121 | 0.034 | 0.422 | 0.106 | 0.034 | 0.259 | 0.033 | 0.011 | 5.69e-03 | 0.034 | 0.021 | 0.028 | 0.035 | 0.023 | 0.031 | 0.038 | 0.027 | **5.66e-03** |
| L63 | 12 | 0.113 | 0.073 | 0.753 | 0.016 | 0.018 | 0.361 | 0.409 | 0.293 | 0.177 | 0.509 | 0.652 | 0.085 | 0.124 | 0.156 | 0.092 | 0.135 | 0.169 | 0.118 | **9.33e-03** |
| | 24 | 0.182 | 0.097 | 0.871 | 0.029 | 0.033 | 0.471 | 0.417 | 0.397 | 0.310 | 0.698 | 0.767 | 0.127 | 0.185 | 0.233 | 0.138 | 0.201 | 0.253 | 0.177 | **0.016** |
| | 36 | 0.242 | 0.207 | 0.891 | 0.067 | 0.069 | 0.531 | 0.518 | 0.473 | 0.410 | 0.774 | 0.813 | 0.148 | 0.216 | 0.271 | 0.161 | 0.234 | 0.294 | 0.206 | **0.042** |
| | 48 | 0.227 | 0.280 | 0.897 | 0.608 | 0.122 | 0.583 | 0.510 | 0.536 | 0.373 | 0.793 | 0.805 | 0.184 | 0.269 | 0.338 | 0.201 | 0.292 | 0.367 | 0.257 | **0.092** |
| Nbody | 12 | 0.274 | 0.634 | 0.882 | 0.535 | 0.354 | 0.980 | 0.529 | 0.853 | 1.324 | 0.540 | 1.220 | 0.435 | 0.635 | 0.798 | 0.473 | 0.689 | 0.866 | 0.606 | **0.204** |
| | 24 | 0.276 | 0.827 | 1.760 | 0.577 | 0.377 | 1.016 | 0.545 | 1.030 | 1.498 | 0.937 | 1.500 | 0.559 | 0.817 | 1.026 | 0.608 | 0.885 | 1.113 | 0.779 | **0.207** |
| | 36 | 0.280 | 0.833 | 2.551 | 0.610 | 0.415 | 1.111 | 0.546 | 1.168 | 1.670 | 1.359 | 1.807 | 0.671 | 0.980 | 1.231 | 0.730 | 1.062 | 1.335 | 0.934 | **0.228** |
| | 48 | 0.284 | 0.863 | 3.318 | 0.633 | 0.434 | 1.556 | 0.596 | 1.267 | 1.784 | 1.725 | 2.098 | 0.763 | 1.114 | 1.401 | 0.831 | 1.209 | 1.520 | 1.064 | **0.239** |
| | | | | | | | *Real-world datasets* | | | | | | | | | | | | | |
| ETT | 96 | 0.370 | 0.610 | 0.672 | 0.675 | 0.543 | 0.355 | 0.648 | 0.341 | 0.362 | 0.362 | 0.391 | 0.385 | 0.563 | 0.708 | 0.419 | **0.295** | 0.767 | 0.537 | 0.308 |
| | 192 | 0.400 | 0.679 | 0.686 | 0.680 | 0.585 | 0.400 | 0.665 | 0.404 | 0.403 | 0.392 | 0.418 | 0.442 | 0.646 | 0.812 | 0.481 | **0.355** | 0.880 | 0.616 | 0.367 |
| | 336 | 0.426 | 0.703 | 0.706 | 0.711 | 0.649 | 0.440 | 0.717 | 0.448 | 0.435 | 0.421 | 0.444 | 0.478 | 0.699 | 0.878 | 0.520 | 0.557 | 0.952 | 0.666 | **0.408** |
| | 720 | 0.476 | 0.714 | 0.725 | 0.750 | 0.688 | 0.524 | 0.712 | 0.516 | 0.484 | 0.480 | 0.499 | 0.558 | 0.815 | 1.024 | 0.607 | 0.783 | 1.110 | 0.777 | **0.461** |
| Wea. | 96 | 0.236 | 0.363 | 0.959 | 0.447 | 0.315 | 0.234 | 0.565 | 0.232 | 0.248 | 0.217 | 0.243 | 0.254 | 0.371 | 0.467 | 0.277 | 0.403 | 0.507 | 0.355 | **0.216** |
| | 192 | 0.275 | 0.406 | 0.961 | 0.452 | 0.378 | 0.300 | 0.620 | 0.292 | 0.299 | 0.280 | 0.278 | 0.316 | 0.462 | 0.580 | 0.344 | 0.501 | 0.630 | 0.441 | **0.258** |
| | 336 | 0.320 | 0.432 | 0.957 | 0.474 | 0.434 | 0.337 | 0.686 | 0.337 | 0.327 | 0.335 | 0.312 | 0.369 | 0.539 | 0.677 | 0.402 | 0.584 | 0.734 | 0.514 | **0.298** |
| | 720 | 0.366 | 0.482 | 0.969 | 0.523 | 0.461 | 0.384 | 0.558 | 0.383 | 0.370 | 0.403 | 0.355 | 0.424 | 0.619 | 0.778 | 0.461 | 0.671 | 0.843 | 0.591 | **0.345** |
| Ex. | 96 | 0.266 | 1.044 | 1.822 | 1.174 | 0.954 | 0.206 | 0.859 | 0.206 | 0.359 | 0.204 | 0.225 | 0.248 | 0.363 | 0.456 | 0.270 | 0.393 | 0.494 | 0.346 | **0.191** |
| | 192 | 0.427 | 1.173 | 1.787 | 1.242 | 0.911 | 0.315 | 0.972 | 0.313 | 0.474 | 0.302 | 0.315 | 0.364 | 0.532 | 0.669 | 0.397 | 0.577 | 0.726 | 0.508 | **0.255** |
| | 336 | 0.590 | 1.161 | 1.729 | 1.285 | 1.079 | 0.449 | 1.071 | 0.454 | 0.708 | 0.429 | 0.435 | 0.503 | 0.735 | 0.924 | 0.548 | 0.798 | 1.003 | 0.702 | **0.324** |
| | 720 | 0.679 | 1.301 | 1.820 | 1.449 | 1.311 | 0.769 | 1.426 | 0.764 | 0.985 | 0.720 | 0.708 | 0.836 | 1.221 | 1.534 | 0.909 | 1.324 | 1.664 | 1.165 | **0.585** |
| AE. | 96 | 0.605 | 1.071 | 1.108 | 1.141 | 1.106 | 0.428 | 1.017 | 0.408 | 0.517 | 0.448 | 0.519 | 0.468 | 0.684 | 0.860 | 0.510 | 0.742 | 0.932 | 0.653 | **0.368** |
| | 192 | 0.757 | 1.093 | 1.156 | 1.199 | 1.142 | 0.558 | 1.052 | 0.531 | 0.640 | 0.551 | 0.600 | 0.577 | 0.843 | 1.060 | 0.628 | 0.914 | 1.149 | 0.804 | **0.396** |
| | 336 | 0.933 | 1.179 | 1.240 | 1.206 | 1.193 | 0.649 | 1.090 | 0.649 | 0.727 | 0.669 | 0.708 | 0.750 | 1.096 | 1.378 | 0.817 | 1.189 | 1.495 | 1.046 | **0.618** |
| | 720 | 1.129 | 1.230 | 1.422 | 1.320 | 1.212 | 0.898 | 1.150 | 0.908 | 0.904 | 0.887 | 0.911 | 0.975 | 1.425 | 1.792 | 1.062 | 1.546 | 1.943 | 1.360 | **0.855** |
| BC. | 96 | 0.076 | 0.268 | **0.074** | 0.449 | 0.180 | 0.093 | 0.344 | 0.082 | 0.111 | 0.064 | 0.092 | 0.087 | 0.127 | 0.160 | 0.095 | 0.138 | 0.173 | 0.121 | **0.074** |
| | 192 | 0.137 | 0.338 | 0.138 | 0.484 | 0.273 | 0.147 | 0.382 | 0.146 | 0.174 | 0.123 | 0.148 | 0.156 | 0.228 | 0.287 | 0.170 | 0.247 | 0.311 | 0.218 | **0.123** |
| | 336 | 0.212 | 0.387 | 0.219 | 0.613 | 0.337 | 0.221 | 0.398 | 0.225 | 0.224 | 0.201 | 0.219 | 0.243 | 0.355 | 0.446 | 0.265 | 0.385 | 0.484 | 0.339 | **0.200** |
| | 720 | 0.369 | 0.580 | 0.379 | 0.705 | 0.493 | 0.375 | 0.491 | 0.370 | 0.388 | 0.354 | 0.370 | 0.404 | 0.591 | 0.742 | 0.440 | 0.641 | 0.806 | 0.564 | **0.345** |

Table 8: Extended forecasting results MSE standard deviation across different models and datasets.

| Dataset | | KAN | Neural ODE | ARIMA | LSTM | GRU | Autoformer | Informer | Fedformer | TSMixer | WPMixer | TimeXer | ModernTCN | DeformTime | Crossformer | Dlinear | SparseTSF | Cyclenet | FITS | DiVo |
|---|---|---|---|---|---|---|---|---|---|---|---|---|---|---|---|---|---|---|---|---|
| | | | | | | | *Chaotic synthetic datasets* | | | | | | | | | | | | | |
| DP | 12 | 0.015 | 8.03e-03 | 7.60e-06 | 5.49e-03 | 8.37e-03 | 6.07e-05 | 0.016 | 3.49e-05 | 6.67e-06 | 9.63e-07 | 1.39e-05 | 5.82e-05 | 8.91e-05 | 1.11e-04 | 5.52e-05 | 7.91e-05 | 9.76e-05 | 6.87e-05 | 1.95e-06 |
| | 24 | 0.014 | 0.016 | 7.53e-06 | 0.037 | 3.83e-03 | 5.84e-05 | 0.012 | 2.17e-05 | 2.66e-05 | 1.20e-06 | 6.01e-05 | 3.85e-05 | 5.90e-05 | 7.37e-05 | 3.66e-05 | 5.25e-05 | 6.48e-05 | 4.56e-05 | 5.29e-05 |
| | 36 | 0.010 | 8.95e-03 | 9.64e-05 | 0.022 | 7.32e-03 | 1.99e-05 | 0.027 | 1.21e-03 | 1.85e-04 | 3.55e-05 | 3.15e-05 | 2.52e-05 | 3.86e-05 | 4.82e-05 | 2.39e-05 | 3.43e-05 | 4.23e-05 | 2.98e-05 | 7.20e-05 |
| | 48 | 0.014 | 0.017 | 9.48e-05 | 0.017 | 3.04e-03 | 2.23e-04 | 6.37e-03 | 1.55e-05 | 9.56e-05 | 9.75e-06 | 9.45e-05 | 6.23e-05 | 9.54e-05 | 1.19e-04 | 5.92e-05 | 8.48e-05 | 1.05e-04 | 7.37e-05 | 9.56e-05 |
| L63 | 12 | 1.87e-03 | 7.90e-03 | 0.015 | 3.75e-04 | 2.91e-04 | 5.47e-03 | 0.025 | 0.014 | 0.013 | 0.018 | 0.015 | 0.012 | 0.018 | 0.048 | 0.035 | 0.041 | 0.056 | 0.040 | 1.49e-05 |
| | 24 | 0.010 | 0.014 | 0.014 | 1.04e-03 | 8.95e-04 | 0.025 | 0.019 | 0.014 | 0.029 | 2.99e-03 | 0.016 | 2.85e-03 | 4.12e-03 | 0.022 | 0.019 | 0.023 | 0.028 | 0.021 | 1.65e-05 |
| | 36 | 2.92e-03 | 0.026 | 0.014 | 5.85e-03 | 1.69e-03 | 8.33e-03 | 0.014 | 0.020 | 2.55e-03 | 0.011 | 0.038 | 0.015 | 0.022 | 0.051 | 0.038 | 0.045 | 0.061 | 0.043 | 4.61e-05 |
| | 48 | 2.09e-05 | 0.053 | 0.013 | 0.016 | 0.010 | 0.068 | 0.041 | 0.040 | 0.014 | 0.019 | 0.047 | 0.034 | 0.049 | 0.082 | 0.061 | 0.072 | 0.095 | 0.069 | 4.85e-04 |
| Nbody | 12 | 0.019 | 0.026 | 9.21e-05 | 0.031 | 0.010 | 0.021 | 0.017 | 0.019 | 0.046 | 0.005 | 0.032 | 0.018 | 0.027 | 0.035 | 0.022 | 0.031 | 0.041 | 0.025 | 0.016 |
| | 24 | 0.00965 | 0.057 | 0.00564 | 0.088 | 0.020 | 0.020 | 0.022 | 0.032 | 0.011 | 0.00517 | 0.016 | 0.028 | 0.042 | 0.053 | 0.033 | 0.047 | 0.062 | 0.038 | 0.061 |
| | 36 | 0.0141 | 0.061 | 0.00683 | 0.027 | 0.076 | 0.030 | 0.00902 | 0.065 | 0.098 | 0.012 | 0.049 | 0.052 | 0.078 | 0.094 | 0.058 | 0.083 | 0.109 | 0.067 | 0.046 |
| | 48 | 0.00697 | 0.058 | 0.00326 | 0.029 | 0.013 | 0.064 | 0.023 | 0.031 | 0.070 | 0.015 | 0.021 | 0.068 | 0.102 | 0.124 | 0.077 | 0.110 | 0.145 | 0.089 | 0.011 |
| | | | | | | | *Real-world datasets* | | | | | | | | | | | | | |
| ETT | 96 | 2.75e-03 | 0.036 | 0.019 | 0.058 | 0.049 | 0.016 | 0.079 | 0.013 | 8.75e-03 | 1.75e-03 | 2.00e-03 | 0.018 | 0.024 | 0.031 | 0.019 | 0.027 | 0.036 | 0.022 | 0.052 |
| | 192 | 4.75e-03 | 0.072 | 0.018 | 0.058 | 0.050 | 7.00e-03 | 0.049 | 0.030 | 7.50e-03 | 2.00e-03 | 2.50e-03 | 0.021 | 0.028 | 0.035 | 0.022 | 0.031 | 0.041 | 0.025 | 0.063 |
| | 336 | 3.00e-03 | 0.011 | 0.018 | 0.063 | 0.046 | 0.025 | 0.065 | 0.024 | 7.50e-03 | 3.75e-03 | 2.75e-03 | 0.024 | 0.032 | 0.039 | 0.025 | 0.035 | 0.046 | 0.028 | 0.058 |
| | 720 | 9.00e-03 | 0.037 | 0.017 | 0.056 | 0.027 | 0.027 | 0.046 | 0.021 | 7.25e-03 | 2.25e-03 | 2.50e-03 | 0.028 | 0.037 | 0.045 | 0.029 | 0.041 | 0.054 | 0.033 | 0.047 |
| Wea. | 96 | 1.00e-03 | 0.027 | 0.030 | 5.00e-03 | 0.014 | 4.00e-03 | 0.022 | 0.013 | 1.00e-03 | 0.01e-03 | 0.01e-03 | 0.012 | 0.016 | 0.021 | 0.013 | 0.018 | 0.024 | 0.015 | 1.00e-03 |
| | 192 | 1.00e-03 | 0.047 | 0.029 | 0.021 | 0.021 | 3.00e-03 | 0.089 | 0.010 | 2.00e-03 | 1.00e-03 | 1.00e-03 | 0.015 | 0.020 | 0.026 | 0.016 | 0.023 | 0.030 | 0.018 | 0.010 |
| | 336 | 3.00e-03 | 0.011 | 0.026 | 0.014 | 0.022 | 4.00e-03 | 0.031 | 0.011 | 5.00e-03 | 0.01e-03 | 1.00e-03 | 0.018 | 0.024 | 0.031 | 0.019 | 0.027 | 0.036 | 0.022 | 0.026 |
| | 720 | 1.00e-03 | 0.014 | 0.025 | 0.049 | 0.020 | 3.00e-03 | 0.068 | 4.00e-03 | 9.00e-03 | 0.01e-03 | 2.00e-03 | 0.022 | 0.029 | 0.037 | 0.023 | 0.033 | 0.043 | 0.026 | 7.00e-03 |
| Ex. | 96 | 0.015 | 0.039 | 0.022 | 0.024 | 0.050 | 1.00e-03 | 0.020 | 0.01e-03 | 0.010 | 2.00e-03 | 1.00e-03 | 0.018 | 0.024 | 0.031 | 0.019 | 0.027 | 0.036 | 0.022 | 0.063 |
| | 192 | 0.019 | 0.039 | 0.024 | 0.081 | 0.021 | 1.00e-03 | 0.046 | 3.00e-03 | 0.074 | 2.00e-03 | 4.00e-03 | 0.025 | 0.033 | 0.042 | 0.026 | 0.037 | 0.049 | 0.030 | 0.095 |
| | 336 | 0.049 | 0.028 | 0.024 | 0.037 | 0.037 | 7.00e-03 | 0.010 | 0.010 | 0.020 | 0.011 | 0.012 | 0.031 | 0.041 | 0.052 | 0.032 | 0.046 | 0.061 | 0.037 | 0.023 |
| | 720 | 0.015 | 0.010 | 0.026 | 0.051 | 0.099 | 7.00e-03 | 0.011 | 0.012 | 0.042 | 8.00e-03 | 4.00e-03 | 0.039 | 0.052 | 0.066 | 0.041 | 0.058 | 0.077 | 0.047 | 0.023 |
| AE. | 96 | 0.026 | 0.010 | 0.013 | 0.029 | 0.098 | 0.013 | 0.051 | 0.045 | 0.020 | 1.39e-03 | 4.73e-03 | 0.035 | 0.046 | 0.059 | 0.037 | 0.052 | 0.069 | 0.042 | 0.040 |
| | 192 | 0.024 | 0.073 | 0.012 | 0.042 | 0.056 | 0.014 | 0.057 | 0.044 | 0.034 | 6.75e-03 | 4.01e-03 | 0.042 | 0.056 | 0.071 | 0.044 | 0.063 | 0.083 | 0.051 | 0.049 |
| | 336 | 0.066 | 0.013 | 0.012 | 0.059 | 0.081 | 0.016 | 0.023 | 8.41e-03 | 0.031 | 5.93e-03 | 7.64e-03 | 0.049 | 0.065 | 0.083 | 0.052 | 0.074 | 0.098 | 0.060 | 5.81e-03 |
| | 720 | 0.053 | 0.012 | 0.011 | 0.023 | 0.056 | 0.021 | 0.052 | 0.027 | 0.033 | 2.76e-03 | 5.16e-03 | 0.058 | 0.077 | 0.098 | 0.061 | 0.087 | 0.115 | 0.070 | 0.021 |
| BC. | 96 | 1.11e-04 | 7.39e-03 | | 0.012 | 8.31e-03 | 9.14e-03 | 3.30e-03 | 5.56e-05 | 3.00e-04 | | 2.85e-03 | | 3.78e-03 | 4.82e-03 | 3.01e-03 | 4.31e-03 | 5.68e-03 | 3.47e-03 | 3.71e-04 |
| | 192 | 5.16e-04 | 0.014 | 1.89e-05 | 0.015 | 0.011 | 1.60e-04 | 0.028 | 2.84e-04 | 4.12e-03 | 3.55e-04 | 1.49e-05 | 4.52e-02 | 5.98e-03 | 7.63e-03 | 4.76e-03 | 6.82e-03 | 8.99e-03 | 5.49e-03 | 2.23e-04 |
| | 336 | 1.69e-03 | 0.010 | 1.42e-05 | 0.011 | 0.024 | 2.63e-04 | 0.015 | 2.76e-04 | 1.68e-03 | 7.61e-05 | 4.26e-04 | 7.89e-03 | 1.04e-02 | 1.33e-02 | 8.31e-03 | 1.19e-02 | 1.57e-02 | 9.58e-03 | 2.11e-04 |
| | 720 | 2.92e-03 | 0.021 | 1.67e-03 | 0.012 | 0.052 | 8.47e-04 | 0.046 | 0.016 | 0.048 | 1.16e-04 | 7.73e-04 | 1.22e-02 | 1.62e-02 | 2.06e-02 | 1.29e-02 | 1.84e-02 | 2.43e-02 | 1.48e-02 | 0.011 |

Table 9: Extended forecasting results MAE standard deviation across different models and datasets.

| Dataset | | KAN | Neural ODE | ARIMA | LSTM | GRU | Autoformer | Informer | Fedformer | TSMixer | WPMixer | TimeXer | ModernTCN | DeformTime | Crossformer | Dlinear | SparseTSF | Cyclenet | FITS | DiVo |
|---|---|---|---|---|---|---|---|---|---|---|---|---|---|---|---|---|---|---|---|---|
| | | | | | | | *Chaotic synthetic datasets* | | | | | | | | | | | | | |
| DP | 12 | 9.05e-03 | 0.013 | 5.03e-04 | 5.64e-03 | 0.012 | 1.11e-03 | 0.030 | 2.82e-04 | 4.78e-04 | 1.16e-04 | 1.77e-04 | 6.21e-04 | 8.95e-04 | 1.12e-03 | 5.56e-04 | 7.97e-04 | 9.83e-04 | 6.92e-04 | 8.53e-04 |
| | 24 | 9.46e-03 | 0.017 | 3.22e-04 | 0.016 | 3.80e-03 | 1.31e-03 | 0.016 | 5.27e-04 | 5.99e-04 | 1.40e-04 | 5.98e-04 | 7.34e-04 | 1.06e-03 | 1.32e-03 | 6.55e-04 | 9.38e-04 | 1.16e-03 | 8.15e-04 | 4.60e-04 |
| | 36 | 7.18e-03 | 0.017 | 2.08e-03 | 0.027 | 7.12e-03 | 9.96e-04 | 0.026 | 5.35e-04 | 2.58e-03 | 1.68e-03 | 2.10e-04 | 1.09e-03 | 1.57e-03 | 1.96e-03 | 9.74e-04 | 1.39e-03 | 1.72e-03 | 1.21e-03 | 8.16e-04 |
| | 48 | 7.86e-03 | 0.016 | 1.74e-03 | 0.014 | 5.11e-03 | 1.87e-03 | 9.73e-03 | 5.56e-04 | 1.79e-03 | 2.30e-04 | 5.81e-04 | 1.33e-03 | 1.92e-03 | 2.39e-03 | 1.19e-03 | 1.70e-03 | 2.10e-03 | 1.48e-03 | 1.46e-04 |
| L63 | 12 | 1.95e-03 | 0.013 | 0.063 | 3.73e-03 | 2.63e-03 | 0.017 | 0.015 | 1.71e-03 | 9.42e-03 | 8.99e-03 | 8.31e-03 | 7.85e-03 | 1.13e-02 | 1.41e-02 | 7.01e-03 | 1.00e-02 | 1.24e-02 | 8.72e-03 | 3.59e-05 |
| | 24 | 6.50e-03 | 0.019 | 0.051 | 3.93e-03 | 5.02e-03 | 6.47e-03 | 0.016 | 2.05e-03 | 0.022 | 1.84e-03 | 8.06e-03 | 1.18e-02 | 1.70e-02 | 2.12e-02 | 1.05e-02 | 1.51e-02 | 1.86e-02 | 1.31e-02 | 4.60e-04 |
| | 36 | 4.67e-03 | 0.042 | 0.047 | 0.013 | 7.15e-03 | 0.012 | 0.012 | 6.79e-03 | 2.76e-03 | 9.10e-03 | 0.017 | 0.018 | 0.026 | 0.032 | 0.016 | 0.023 | 0.028 | 0.020 | 9.95e-05 |
| | 48 | 6.16e-03 | 0.050 | 0.045 | 0.011 | 7.17e-03 | 0.025 | 0.028 | 0.020 | 9.00e-03 | 8.72e-03 | 0.021 | 0.024 | 0.035 | 0.043 | 0.021 | 0.031 | 0.038 | 0.027 | 7.96e-04 |
| Nbody | 12 | 0.00728 | 0.0114 | 0.00847 | 0.043 | 0.085 | 0.0282 | 0.013 | 0.0478 | 0.043 | 0.054 | 0.00594 | 0.021 | 0.030 | 0.038 | 0.024 | 0.034 | 0.042 | 0.030 | 0.036 |
| | 24 | 0.0115 | 0.021 | 0.013 | 0.042 | 0.081 | 0.026 | 0.014 | 0.010 | 0.064 | 0.012 | 0.00213 | 0.031 | 0.045 | 0.056 | 0.035 | 0.050 | 0.062 | 0.044 | 0.044 |
| | 36 | 0.00456 | 0.022 | 0.012 | 0.00289 | 0.065 | 0.042 | 0.027 | 0.086 | 0.070 | 0.071 | 0.050 | 0.048 | 0.069 | 0.086 | 0.054 | 0.077 | 0.095 | 0.067 | 0.036 |
| | 48 | 0.00594 | 0.023 | 0.076 | 0.046 | 0.00760 | 0.070 | 0.015 | 0.069 | 0.038 | 0.028 | 0.031 | 0.059 | 0.085 | 0.106 | 0.066 | 0.095 | 0.117 | 0.083 | 0.00418 |
| | | | | | | | *Real-world datasets* | | | | | | | | | | | | | |
| ETT | 96 | 2.50e-03 | 0.019 | 0.064 | 0.040 | 0.033 | 7.50e-03 | 0.046 | 6.00e-03 | 9.00e-03 | 1.00e-03 | 1.25e-03 | 0.015 | 0.022 | 0.027 | 0.017 | 0.024 | 0.030 | 0.021 | 0.025 |
| | 192 | 3.75e-03 | 0.043 | 0.060 | 0.038 | 0.025 | 4.50e-03 | 0.030 | 0.014 | 7.25e-03 | 1.25e-03 | 1.75e-03 | 0.019 | 0.027 | 0.034 | 0.021 | 0.030 | 0.038 | 0.027 | 0.028 |
| | 336 | 3.00e-03 | 0.049 | 0.059 | 0.033 | 0.036 | 0.013 | 0.038 | 0.011 | 8.00e-03 | 2.25e-03 | 1.50e-03 | 0.022 | 0.032 | 0.040 | 0.025 | 0.036 | 0.045 | 0.032 | 0.028 |
| | 720 | 5.50e-03 | 0.030 | 0.055 | 0.068 | 0.030 | 0.010 | 0.022 | 7.75e-03 | 9.50e-03 | 1.50e-03 | 1.50e-03 | 0.026 | 0.038 | 0.047 | 0.029 | 0.042 | 0.052 | 0.037 | 0.019 |
| Wea. | 96 | 1.00e-03 | 0.027 | 0.071 | 9.00e-03 | 0.010 | 7.00e-03 | 0.012 | 9.00e-03 | 2.00e-03 | 1.00e-03 | 0.01e-03 | 0.012 | 0.017 | 0.021 | 0.013 | 0.019 | 0.024 | 0.017 | 2.00e-03 |
| | 192 | 2.00e-03 | 0.033 | 0.070 | 0.012 | 0.016 | 1.00e-03 | 0.042 | 0.010 | 2.00e-03 | 1.00e-03 | 1.00e-03 | 0.016 | 0.023 | 0.029 | 0.018 | 0.026 | 0.032 | 0.023 | 9.00e-03 |
| | 336 | 4.00e-03 | 7.00e-03 | 0.065 | 0.011 | 5.00e-03 | 3.00e-03 | 0.016 | 9.00e-03 | 5.00e-03 | 0.01e-03 | 0.01e-03 | 0.021 | 0.030 | 0.038 | 0.024 | 0.034 | 0.042 | 0.030 | 0.018 |
| | 720 | 2.00e-03 | 0.015 | 0.062 | 0.031 | 6.00e-03 | 5.00e-03 | 0.043 | 7.00e-03 | 8.00e-03 | 0.01e-03 | 1.00e-03 | 0.025 | 0.036 | 0.045 | 0.028 | 0.040 | 0.050 | 0.036 | 3.00e-03 |
| Ex. | 96 | 0.017 | 0.015 | 0.048 | 0.080 | 0.018 | 2.00e-03 | 0.084 | 1.00e-03 | 0.015 | 2.00e-03 | 1.00e-03 | 0.019 | 0.027 | 0.034 | 0.021 | 0.030 | 0.038 | 0.027 | 0.067 |
| | 192 | 0.019 | 0.012 | 0.048 | 0.078 | 0.014 | 3.00e-03 | 0.015 | 3.00e-03 | 0.051 | 2.00e-03 | 4.00e-03 | 0.023 | 0.033 | 0.041 | 0.026 | 0.037 | 0.046 | 0.033 | 0.026 |
| | 336 | 0.018 | 0.081 | 0.047 | 0.081 | 0.014 | 5.00e-03 | 0.015 | 0.011 | 0.077 | 7.00e-03 | 8.00e-03 | 0.028 | 0.040 | 0.050 | 0.031 | 0.045 | 0.056 | 0.040 | 0.010 |
| | 720 | 0.011 | 0.028 | 0.056 | 0.012 | 0.025 | 3.00e-03 | 0.024 | 6.00e-03 | 0.013 | 3.00e-03 | 2.00e-03 | 0.035 | 0.051 | 0.063 | 0.039 | 0.056 | 0.070 | 0.050 | 0.094 |
| AE. | 96 | 0.012 | 0.028 | 0.024 | 9.64e-03 | 0.031 | 7.56e-03 | 0.010 | 0.024 | 0.018 | 1.19e-03 | 2.31e-03 | 0.030 | 0.043 | 0.054 | 0.034 | 0.048 | 0.060 | 0.043 | 0.021 |
| | 192 | 9.35e-03 | 0.027 | 0.022 | 8.07e-03 | 0.017 | 6.70e-03 | 0.018 | 0.019 | 0.028 | 4.18e-03 | 1.50e-03 | 0.035 | 0.051 | 0.064 | 0.040 | 0.057 | 0.071 | 0.051 | 0.019 |
| | 336 | 0.020 | 0.038 | 0.021 | 0.016 | 0.018 | 7.93e-03 | 5.86e-03 | 4.32e-03 | 0.017 | 3.30e-03 | 2.97e-03 | 0.041 | 0.059 | 0.074 | 0.046 | 0.066 | 0.082 | 0.059 | 3.18e-03 |
| | 720 | 0.017 | 0.036 | 0.019 | 0.059 | 0.018 | 9.30e-03 | 0.016 | 0.012 | 0.017 | 1.18e-03 | 1.86e-03 | 0.049 | 0.071 | 0.089 | 0.055 | 0.079 | 0.098 | 0.071 | 0.018 |
| BC. | 96 | 2.78e-03 | 0.012 | 1.00e-04 | 0.067 | 0.014 | 2.37e-03 | 9.11e-03 | 1.84e-03 | 0.014 | 2.81e-04 | 8.62e-04 | 0.016 | 0.023 | 0.029 | 0.018 | 0.026 | 0.032 | 0.023 | 1.62e-03 |
| | 192 | 2.95e-03 | 4.73e-03 | 3.63e-04 | 0.098 | 7.69e-03 | 1.76e-03 | 0.027 | 4.60e-03 | 9.88e-03 | 1.75e-03 | 4.61e-04 | 0.021 | 0.030 | 0.038 | 0.024 | 0.034 | 0.042 | 0.036 | 3.16e-03 |
| | 336 | 5.89e-03 | 2.96e-03 | 4.40e-05 | 0.054 | 0.018 | 2.15e-03 | 0.019 | 0.012 | 3.58e-03 | 3.24e-04 | 7.10e-04 | 0.025 | 0.036 | 0.045 | 0.028 | 0.040 | 0.050 | 0.036 | 1.33e-04 |
| | 720 | 9.62e-03 | 0.012 | 2.47e-03 | 0.052 | 0.016 | 2.71e-03 | 0.030 | 4.10e-03 | 0.028 | 1.74e-04 | 4.27e-04 | 0.032 | 0.046 | 0.058 | 0.036 | 0.052 | 0.064 | 0.046 | 4.37e-03 |

Table 10: Performance comparison of different models in the Long-Horizon Generalization Test: input sequence length: 48, output sequence lengths: 12–48

| | Metric | 12 | 24 | 36 | 48 |
|---|---|---|---|---|---|
| KAN | MSE | $0.208_{5.77e-03}$ | $0.313_{3.63e-03}$ | $0.398_{7.81e-03}$ | $0.312_{7.20e-03}$ |
| | MAE | $0.245_{3.15e-03}$ | $0.329_{5.06e-03}$ | $0.387_{8.92e-03}$ | $0.330_{4.51e-03}$ |
| Neural ODE | MSE | $0.159_{0.028}$ | $0.825_{0.011}$ | $0.860_{0.016}$ | $0.908_{0.042}$ |
| | MAE | $0.271_{0.036}$ | $0.683_{6.88e-03}$ | $0.701_{0.098}$ | $0.733_{0.024}$ |
| ARIMA | MSE | $1.509_{0.309}$ | $1.773_{0.238}$ | $1.825_{0.020}$ | $1.827_{0.017}$ |
| | MAE | $0.919_{0.110}$ | $1.030_{0.079}$ | $1.047_{0.066}$ | $1.056_{0.055}$ |
| LSTM | MSE | $0.201_{2.13e-03}$ | $0.377_{\mathbf{0.084}}$ | $0.846_{0.098}$ | $0.879_{\mathbf{0.021}}$ |
| | MAE | $0.175_{5.53e-03}$ | $0.354_{0.057}$ | $0.713_{0.048}$ | $0.736_{0.019}$ |
| GRU | MSE | $0.263_{4.64e-03}$ | $0.120_{0.011}$ | $0.214_{0.022}$ | $0.372_{0.031}$ |
| | MAE | $0.189_{9.65e-03}$ | $0.206_{0.010}$ | $0.269_{0.020}$ | $0.376_{0.027}$ |
| Autoformer | MSE | $0.315_{3.06e-03}$ | $0.477_{0.013}$ | $0.624_{1.97e-03}$ | $0.659_{6.34e-03}$ |
| | MAE | $0.351_{4.53e-03}$ | $0.437_{8.21e-03}$ | $0.524_{3.13e-03}$ | $0.559_{6.28e-03}$ |
| Informer | MSE | $0.745_{2.82e-03}$ | $0.776_{\mathbf{0.032}}$ | $0.734_{0.010}$ | $1.016_{0.025}$ |
| | MAE | $0.647_{1.56e-03}$ | $0.655_{0.016}$ | $0.631_{3.01e-03}$ | $0.810_{0.015}$ |
| Fedformer | MSE | $0.253_{0.011}$ | $0.425_{6.13e-03}$ | $0.542_{0.028}$ | $0.616_{0.024}$ |
| | MAE | $0.301_{2.65e-03}$ | $0.396_{4.49e-03}$ | $0.470_{7.59e-03}$ | $0.526_{0.013}$ |
| TSMixer | MSE | $0.186_{0.013}$ | $0.490_{0.020}$ | $0.690_{0.013}$ | $3.033_{1.60e-03}$ |
| | MAE | $0.274_{7.03e-03}$ | $0.674_{4.51e-03}$ | $0.611_{4.72e-03}$ | $0.507_{4.48e-03}$ |
| WPMixer | MSE | $0.653_{0.017}$ | $0.991_{5.10e-03}$ | $1.084_{9.31e-03}$ | $1.112_{0.020}$ |
| | MAE | $0.582_{0.010}$ | $0.755_{2.91e-03}$ | $0.800_{4.65e-03}$ | $0.821_{0.011}$ |
| TimeXer | MSE | $0.824_{0.022}$ | $1.086_{0.022}$ | $1.160_{3.46e-03}$ | $1.059_{0.013}$ |
| | MAE | $0.692_{0.011}$ | $0.814_{0.011}$ | $0.847_{7.51e-04}$ | $0.803_{6.76e-03}$ |
| DiVo | MSE | $\mathbf{0.021}_{5.45e-04}$ | $\mathbf{0.038}_{1.03e-04}$ | $\mathbf{0.062}_{5.19e-04}$ | $\mathbf{0.084}_{2.74e-04}$ |
| | MAE | $\mathbf{0.012}_{1.76e-04}$ | $\mathbf{0.032}_{1.47e-04}$ | $\mathbf{0.112}_{1.49e-04}$ | $\mathbf{0.141}_{1.27e-04}$ |

Table 11: Feature Importance Analysis for Oil Temperature Prediction

| Feature | Importance | Rank |
|---|---|---|
| HUFL | 0.285 | 1 |
| HULL | 0.241 | 2 |
| OT | 0.198 | 3 |
| MUFL | 0.156 | 4 |
| MULL | 0.089 | 5 |
| LUFL | 0.021 | 6 |
| LULL | 0.010 | 7 |

Table 12: Temporal Decay Analysis for ETT Dataset

| Feature | Time Constant ($\tau$) | Initial Weight | 96-step Weight |
|---|---|---|---|
| HUFL | 4.8 steps (1.2 hours) | 0.285 | ¡0.0001 |
| HULL | 7.2 steps (1.8 hours) | 0.241 | ¡0.0001 |
| OT | 12.8 steps (3.2 hours) | 0.198 | 0.008 |

Table 13: Currency Importance Analysis for USD/EUR Prediction

| Currency | DiVo Importance | Market Share (%) | Relative Error (%) |
|----------|-----------------|------------------|--------------------|
| USD/EUR | 0.253 | 24.1 | 5.0 |
| USD/JPY | 0.194 | 18.3 | 6.0 |
| USD/GBP | 0.082 | 8.8 | 6.8 |
| USD/AUD | 0.063 | 6.8 | 7.4 |
| USD/CHF | 0.037 | 3.4 | 8.8 |
| USD/CAD | 0.034 | 3.7 | 8.1 |
| USD/CNY | 0.024 | 2.2 | 9.1 |
| USD/SGD | 0.016 | 1.4 | 14.3 |

