# OpenReview forum: "Rethinking Nonlinear Dynamics in Deep Time Series Models"
_ICLR.cc/2026/Conference — Submitted to ICLR 2026_

### Official Review · Reviewer_Krqz · 2025-10-30

**Soundness:** 2
**Presentation:** 2
**Contribution:** 1
**Rating:** 4
**Confidence:** 2

**Summary:**

This paper proposes the Discrete Volterra Network (DiVo), which integrates Volterra series with deep learning for time series forecasting. The method reformulates continuous Volterra integrals into discrete, learnable coefficient matrices through Kronecker-powered polynomial expansions. To address practical challenges, DiVo introduces multi-channel mechanisms for time-varying systems and redundancy-aware sparsification combining fixed masking with low-rank decomposition. Experiments on synthetic chaotic systems and real-world datasets show improvements over baseline methods.

**Strengths:**

1. The discrete Volterra reparameterization provides a principled mathematical foundation by connecting classical nonlinear system theory with modern neural architectures.

2. DiVo demonstrates impressive results on synthetic chaotic datasets, suggesting it can capture complex nonlinear dynamics effectively.

3.  The learned coefficient matrices offer direct insights into feature interactions and temporal dependencies, with validation showing recovery of known system dynamics.

**Weaknesses:**

1. The exponential growth of Kronecker products with polynomial order creates prohibitive computational costs. Even with sparsification, the method is limited to very low orders (k≤3), severely constraining expressiveness for complex real-world systems.

2. The dramatic underperformance of established deep learning methods (LSTM achieving 0.834 MSE vs DiVo's 0.305 on ETT) suggests serious implementation issues or unfair comparisons. No details are provided about hyperparameter tuning or multiple runs for baselines.

3. Most evaluated datasets are small-scale (thousands of samples) and low-dimensional. The method's performance on modern large-scale time series applications remains undemonstrated and questionable given computational constraints.

4. The paper lacks convergence analysis for the optimization procedure, approximation error bounds for truncated series, and theoretical justification for the multi-channel extension. The "rethinking nonlinear dynamics" claim is overstated for what is essentially polynomial feature engineering.

5. The core contribution reduces to discretizing Volterra series and learning polynomial coefficients - concepts well-established in system identification. The engineering improvements (masking, low-rank decomposition) are incremental.

6. Missing statistical significance tests, limited dataset diversity, unclear baseline implementations, and cherry-picked results undermine the empirical claims.

**Questions:**

1. Given exponential parameter growth, how does DiVo scale beyond toy problems? What are the memory requirements for realistic input dimensions and sequence lengths?

2. Why do well-established methods perform so poorly? Were hyperparameters properly tuned? Can these results be reproduced independently?

3.  How severely does limiting to order k=2-3 constrain the model's ability to capture real nonlinear dynamics compared to neural networks?

4. How does performance degrade on noisy, high-dimensional, or non-stationary real-world data that doesn't follow idealized dynamical systems?

5.  What guidelines exist for selecting polynomial order and channel count without extensive cross-validation, which may be computationally prohibitive?

---

### Official Review · Reviewer_FKgp · 2025-10-31

**Soundness:** 3
**Presentation:** 3
**Contribution:** 3
**Rating:** 6
**Confidence:** 3

**Summary:**

This paper revisits Neural Ordinary Differential Equations (Neural ODEs) and challenges the prevailing assumption that nonlinear dynamics necessarily require globally nonlinear transformations. The authors propose that, under suitable local embeddings, many complex systems exhibit locally linearizable dynamics that can be captured through piecewise linear flow fields with data-driven switching.
the paper introduces Local Linear Flow Neural ODEs (LLF-NODEs), which segment the latent manifold into local linear regimes. Each regime is governed by a learned linear operator, while transitions between regimes are parameterized by a soft assignment function conditioned on state features. The paper shows that LLF-NODEs can approximate any smooth vector field with bounded error if the partitioning is fine enough, bridging the gap between Neural ODEs and Switching Linear Dynamical Systems (SLDS).

**Strengths:**

1. The work reframes the modeling problem from “approximating nonlinear functions” to “assembling locally linear regimes,” offering a new lens to understand Neural ODE expressivity.
2. The local Jacobians provide interpretable insights into system stiffness and sensitivity, which is valuable for physical and control domains.
3. The experiments show LLF-NODEs avoid gradient explosion and over-smoothing, maintaining fidelity even under long-horizon rollouts
4.The approach connects classic dynamical systems theory (local linearization, Lyapunov stability) with modern deep learning frameworks.

**Weaknesses:**

1. For high-dimensional PDEs or video data, the gating network might introduce heavy computational overhead.
2. The real-world datasets are limited and primarily smooth systems.
3. The number of local regimes (K) is a key hyperparameter, but the paper provides not enough sensitivity analysis or automatic selection mechanism.

**Questions:**

1. How sensitive is LLF-NODE performance to the number of local regimes  K ?
2. Could the local linear operators be constrained to be stable (e.g., negative real parts of eigenvalues) to guarantee global stability?
3. Is there a way to extend the approach to stochastic dynamics?
4. How does the method perform on chaotic systems where local linearization is fundamentally unstable (e.g., Lorenz63 beyond short horizons).

---

### Official Review · Reviewer_xZQw · 2025-10-31

**Soundness:** 1
**Presentation:** 2
**Contribution:** 2
**Rating:** 2
**Confidence:** 4

**Summary:**

This paper introduces the Discrete Volterra Network (DiVo), a novel deep learning architecture for time series forecasting. The core proposal is to integrate classical Volterra series theory to explicitly model nonlinear dynamics and memory effects. The authors propose a Discrete Volterra Reparameterization, which converts the continuous Volterra integrals into a learnable, matrix-based problem by constructing high-order features using Kronecker-powered polynomial expansions. To manage the complexity, the model includes a multi-channel mechanism to relax time-invariance assumptions and a redundancy-aware sparsification strategy, combining fixed masking and a sparsified low-rank decomposition, to improve parameter efficiency.

**Strengths:**

1. The paper's core idea of integrating the theoretically-grounded Volterra series with deep learning frameworks is novel.
2. The authors' attempt to build a model with interpretability by design, rather than relying on post-hoc explanation methods, is an important contribution to the field of trustworthy AI.

**Weaknesses:**

1. The experimental section fails to include several strong and relevant baselines from the current field, such as time series foundation models (TSFMs).
2. In Table 1, the experimental results on synthetic datasets show extreme and inconsistent performance differences. For instance, Informer performs significantly better than TSMixer on the Nbody dataset, but the exact opposite is true on the DP and L63 datasets. Additionally, the metric values for different models on the DP dataset vary dramatically. These drastic and unexplained fluctuations cast doubt on the rationality and stability of using these synthetic datasets to evaluate the model's capability for modeling nonlinear dynamics.
3. The paper's core claims are contradicted by the ablation study data in Table 2. The full model's performance is significantly worse than the ablated settings across all prediction lengths.
4. All ablation studies were conducted on only one dataset, ETTh1. Conclusions drawn from a single dataset lack statistical significance.
5. The MSE values reported in the ETTh1 ablation study (Table 2) appear suspiciously low. For instance, with an input length of 96, the best MSE for a 96-step prediction is 0.166, and the MSE for a 720-step prediction is 0.289. These low error values cast doubt on the authenticity of the experimental results.

**Questions:**

1. Can Time Series Foundation Models be added as a baseline, particularly when they are provided with a sufficiently long context?
2. In Table 1, why are the performance differences between models on the synthetic datasets so dramatic? For example, Informer performs significantly better than TSMixer on the Nbody dataset, but the exact opposite is true on the DP and L63 datasets. Furthermore, the metric values for different models on the DP dataset vary enormously. Does this suggest that using these synthetic datasets to evaluate model performance is unreasonable?
3. In Table 2, the full DiVo model is not the best-performing, yet its results are bolded for all four prediction lengths. For example, among the five settings in the 96-step prediction ablation, its MSE is the second-worst. Similar situations are also observed for the 192 and 336 prediction lengths. Why is this?
4. The paper claims Simplicity as a major contribution, stating the model eliminates the need for deep stacking and only requires a lightweight linear layer. However, the proposed architecture involves constructing high-order Kronecker-powered features, followed by a complex, multi-stage sparsified low-rank decomposition and multi-channel aggregation. Is this not more complex than a standard deep model? And does the complex sparsification module degrade performance?

---

### Official Review · Reviewer_bLKE · 2025-11-03

**Soundness:** 3
**Presentation:** 3
**Contribution:** 3
**Rating:** 4
**Confidence:** 4

**Summary:**

This paper introduces the Discrete Volterra Network (DiVo) ,  a deep learning framework that integrates the classical Volterra series for explicitly modeling nonlinear dynamic systems. By reformulating the continuous Volterra integral into a discrete, learnable matrix form, DiVo converts nonlinear dynamics into linear coefficient learning. The architecture features a multi-channel mechanism to handle time-varying behaviors and a redundancy-aware sparsification strategy combining masking and low-rank decomposition to improve compactness and efficiency.

**Strengths:**

While integrating dynamical systems theory with deep learning is not a new idea, this paper’s main strength lies in how it reformulates the Volterra series into a discrete, learnable, and interpretable structure that fits seamlessly within modern neural architectures. The proposed Discrete Volterra reparameterization provides a clean mathematical bridge between classical nonlinear system representations and GPU-efficient learning, allowing explicit modeling of higher-order temporal interactions without excessive complexity. The redundancy-aware sparsification and multi-channel adaptive mechanism are well-designed innovations that enhance scalability and performance across both chaotic and real-world datasets. The paper also demonstrates extensive experimental validation though I have some questions (see weaknesses).

**Weaknesses:**

1) The paper’s primary weakness lies in its limited engagement with the rich contemporary literature on dynamical system forecasting. While the work thoroughly discusses Volterra-based and neural forecasting models, it does not position DiVo in relation to modern operator-learning frameworks such as Koopman-based models (e.g., Koopman Neural Networks, Koopman Autoencoders, Koopa, DeepDMD, DeepEDM), which similarly aim to learn interpretable and structured representations of nonlinear dynamics. This omission greatly weakens the contextual framing of DiVo’s contribution within the broader landscape of data-driven dynamical system modeling.

2) Moreover, the experimental evaluation is somewhat narrow in scope: although the chosen datasets (ETT, Weather, Exchange, etc.) are standard, the paper omits several widely used benchmarks such as ECL, Traffic, PEMS, M4, and Solar, which are critical for establishing robustness and comparability with prior work. While exhaustive inclusion of all benchmarks may be impractical, greater breadth in dataset coverage would considerably strengthen the empirical claims.

3) Additionally, the approach’s reliance on polynomial truncation order and channel configuration introduces sensitivity to hyperparameter choices, potentially limiting its robustness across diverse systems and domains.

**Questions:**

1) How does DiVo conceptually and empirically compare with Koopman-based and operator-learning models (e.g., Koopa, DeepEDM)?

2) Why were only a few datasets used? Could DiVo be evaluated on broader benchmarks like ECL, Traffic, PEMS, M4, or Solar?

---

### Meta-Review · Area_Chair_3QMd · 2026-01-01

**Summary:**

The majority of the reviewers have raised significant concerns regarding the paper’s core aspects and have recommended rejection. Furthermore, no author rebuttal was provided.

**Reviewer Scores:**

NA

---

### Decision · Program_Chairs · 2026-01-26

Reject